# SDTagNet: Leveraging Text-Annotated Navigation Maps for Online HD Map Construction

**Fabian Immel**[1]    **Jan-Hendrik Pauls**[2]    **Richard Fehler**[1]    **Frank Bieder**[1]
**Jonas Merkert**[2]    **Christoph Stiller**[2]

[1]FZI Research Center for Information Technology    [2]Karlsruhe Institute of Technology
{immel, fehler, bieder}@fzi.de   {jan-hendrik.pauls, stiller}@kit.edu

## Abstract

Autonomous vehicles rely on detailed and accurate environmental information to operate safely. High definition (HD) maps offer a promising solution, but their high maintenance cost poses a significant barrier to scalable deployment. This challenge is addressed by online HD map construction methods, which generate local HD maps from live sensor data. However, these methods are inherently limited by the short perception range of onboard sensors. To overcome this limitation and improve general performance, recent approaches have explored the use of standard definition (SD) maps as prior, which are significantly easier to maintain. We propose SDTagNet, the first online HD map construction method that fully utilizes the information of widely available SD maps, like OpenStreetMap, to enhance far range detection accuracy. Our approach introduces two key innovations. First, in contrast to previous work, we incorporate not only polyline SD map data with manually selected classes, but additional semantic information in the form of textual annotations. In this way, we enrich SD vector map tokens with NLP-derived features, eliminating the dependency on predefined specifications or exhaustive class taxonomies. Second, we introduce a point-level SD map encoder together with orthogonal element identifiers to uniformly integrate all types of map elements. Experiments on Argoverse 2 and nuScenes show that this boosts map perception performance by up to +5.9 mAP (+45%) w.r.t. map construction without priors and up to +3.2 mAP (+20%) w.r.t. previous approaches that already use SD map priors. https://github.com/immel-f/SDTagNet

## 1 Introduction

Autonomous vehicles require a variety of information about their static environment to drive safely. This information is usually provided in form of high definition (HD) maps that contain highly accurate and detailed lane-level road geometry, traffic lights, road signs, traffic rules, *etc*. The extension of self-driving fleets to larger and larger areas showed that the creation and maintenance of HD maps requires a huge amount of computational and manual effort, hence, posing a major obstacle for scalable autonomous driving. To overcome this issue, a variety of approaches have proposed to construct or perceive HD maps online from onboard sensor data [13, 15, 16].

Online HD map construction has the downside of being limited by the range and coverage of onboard sensors. A variety of prior information and representations can be used to mitigate this restriction and improve performance especially in the long range and for occluded areas. These priors can include data from previously visited scenes [26, 32, 23, 27] as well as data from existing prior maps [24, 8, 9, 21, 28]. This prior map data is usually provided in the form of dense grids, sparse vectorized graph representations, or both [31] with different levels of richness.

39th Conference on Neural Information Processing Systems (NeurIPS 2025).

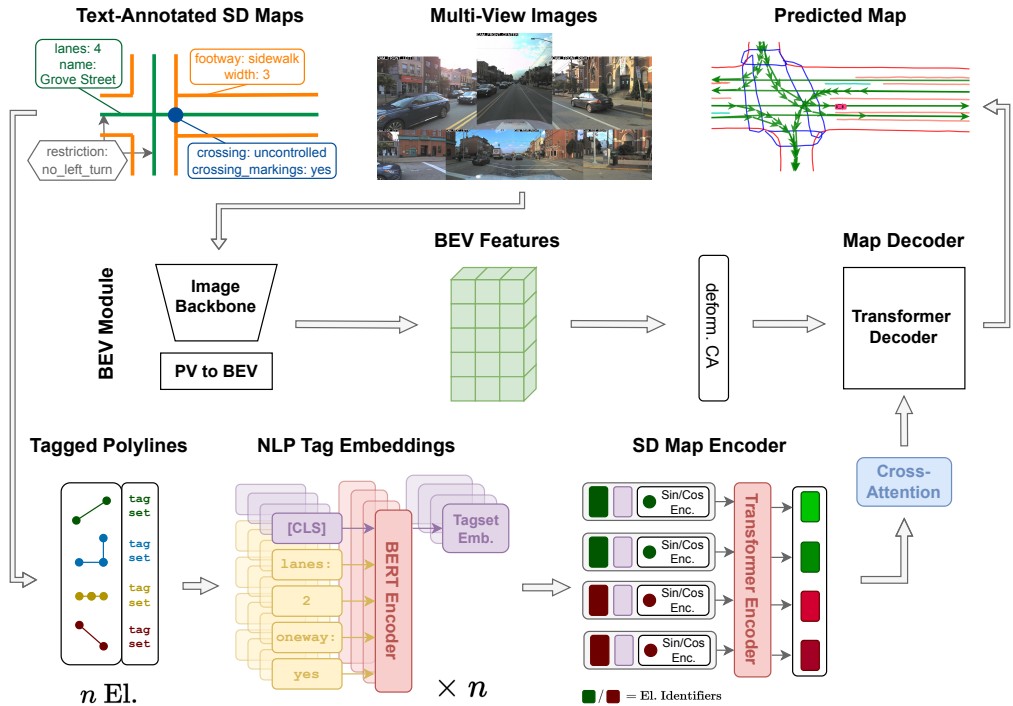

Figure 1: Overview of the model architecture of SDTagNet. To fully exploit textual annotations and all element types in large public SD map databases like OpenStreetMap [22], SDTagNet introduces novel NLP tag embedding and SD map encoder modules. Text annotation embeddings are first computed with a BERT [5] embedding model. They are then fused with scene-level context in a SD map encoder, which uses graph transformer-like methods to flexibly encode points, polylines and element relations. The encoded information is finally supplied to the base model via cross-attention.

Using HD maps as priors only partially alleviates the issues of their creation and maintenance and raises new issues like map change detection [12]. In contrast, standard definition (SD) maps, currently used for navigation and routing, have a far lower demand on accuracy and level of detail, hence, reducing the maintenance effort by orders of magnitude. This motivates the use of SD maps as prior for online HD map construction. With OpenStreetMap [22], crowd-sourced SD map data is freely available on a global scale and kept up-to-date by a community of volunteers.

As we show in this work, existing approaches for incorporating SD map priors do not fully utilize all valuable data contained in OpenStreetMap, but are restricted to dense or sparse representations of a subset of so-called *ways*, *i.e.* polylines of roads and streets, associated with only a few selected tagged text attributes [9, 21, 31]. While these selective, manual SD map features already show great improvements, OpenStreetMap also contains point-level *nodes*, abstract *relations*, and a huge amount of additional textual information annotated to its map elements. To the best of our knowledge, this majority of SD map data has not yet been used to boost online HD map construction performance.

**Contributions** We present SDTagNet, an SD map prior encoding module that can fully utilize *all* information in SD maps, including text annotations, without manual feature engineering. Specifically:

- We increase the SD map element encoder query resolution from polyline-level to point-level, allowing for greater expressiveness by additional integration of point map elements.

- We complement it with explicit orthogonal random feature element identifiers used in graph transformers [10], effectively encoding points, polylines and relations in a joint fashion.

- We propose to use a BERT [5] natural language processing (NLP) encoder and contrastive pretraining to create NLP tag embeddings and hence utilize the entirety of textual annotations in addition to geometric element features in an end-to-end trainable manner.

- Evaluation on Argoverse 2 and nuScenes shows that SDTagNet outperforms existing SD map prior encoding modules by up to 20% and 35%, respectively, especially at far range.

## 2   Related Work

**Online HD Map Construction**   HDMapNet [13] pioneered in defining the task of online HD map construction, *i.e.* to predict a set of vectorized map instances represented as polygons and polylines in a 2D BEV grid, and proposed to solve it by inferring from intermediate semantic segmentation outputs in a heuristic postprocessing step. Subsequent works, including VectorMapNet [19] and MapTR [15], transitioned from this multi-stage pipeline to an end-to-end formulation using DETR-style transformers [2] for direct vectorized map instance prediction. While VectorMapNet adopts an autoregressive decoding strategy, MapTR accelerates inference through a hierarchical bipartite matching mechanism with a fixed number of points per instance. MapTRv2 [16] further advances the architecture by introducing auxiliary supervision, reformulating the point-to-point matching process, and integrating a one-to-many query design to improve both detection performance and convergence. It has since become a baseline architecture for many follow-up research directions [3, 4, 33].

**Priors for Online HD Map Construction**   Recent work seeks to improve detection quality and range by integrating information from previously recorded scenes [26, 32, 23, 27] or existing HD and SD maps [24, 8, 9, 21, 28]. Information from previously visited scenes can either consist of recently recorded temporal information, *e.g.* in [3, 30], or maintained and updated in a global map prior data base [26]. In addition to temporal memory, many real-world autonomous systems have access to pre-built maps ranging from SD maps to HD maps for tasks such as path planning [6]. Recent online map prediction approaches incorporate these as priors to complement incomplete or outdated maps [24, 8]. Among early works, MapEX [24] introduced a mechanism to directly fuse existing HD map elements as embedded query tokens with classic learned queries and online sensor data. M3TR [8] continues this line of work by exploiting query embeddings and training regimes to incorporate semantically diverse priors resulting from varying map degradation scenarios. Concerning SD maps, PMapNet [9] rasterizes OpenStreetMap (OSM) [22] road graphs into images and aims to predict an HD map around it with online sensor information and a pre-trained HD map prior module trained to capture the distribution and structure of the target HD map representation. In contrast, SMERF [21] converts OSM-based SD maps into a sequence of polyline instances representing a fixed set of seven semantic road types and encodes them with a transformer module. Furthermore, TopoSD [28] proposes a hybrid approach, encoding existing SD maps into both 2D feature grids and a set of vectorized instance tokens. As in SMERF [21], these are fused with BEV features from sensor data using a BEVFormer-style encoder [14] with cross-attention. Despite these advances, all of the above use only a subset of available SD map content by hand-selecting attributes.

## 3   Method

An overview of SDTagNet is shown in Figure 1. Beginning with an introduction of annotations in SD maps (Section 3.1), we describe the two main components of SDTagNet: The NLP tag embedding module (Section 3.2) and the SD map encoder (Section 3.3).

### 3.1   Annotations in SD Maps

A key challenge in leveraging standard-definition (SD) maps for online HD map construction lies in understanding the spectrum between SD map and HD map representations. This distinction can be broken down into three main aspects. First, SD maps are less accurate and have lower resolution compared to HD maps, usually in the range of meters compared to centimeters. Like other works in this area [9, 21], we assume that the SD map is roughly localized close to the target HD map and let remaining inaccuracies be compensated by the network during training.

Second, SD maps are often considered sparse in content, with previous approaches only considering roads, *e.g.* by filtering for the `highway` tag in OpenStreetMap (OSM). However, as shown in Figure 2, SD maps like those widely available from the OSM [22] project contain many more kinds of elements. They contain points like traffic lights, bus stops, traffic signs, *etc.* as well as many ways beyond

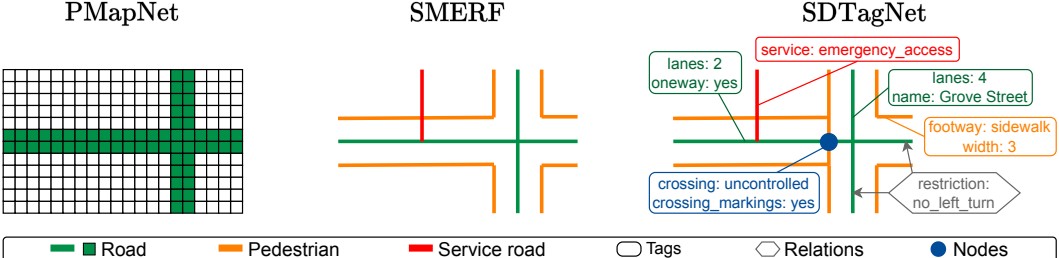

Figure 2: Visualization of the SD map prior input data utilized by existing methods. Existing approaches are limited to rasterized images or polylines with manually defined classes. SDTagNet is the first method that can handle open-vocabulary textual annotations and diverse element types such as points, polylines, and relational information.

roads, like pedestrian paths, building borders, *etc*. Virtual elements, so-called relations, even denote semantic or topological links between two elements, *e.g*. that no left turns are allowed.

Finally, there is a wide variety of descriptive text information annotated to many of those elements. This valuable knowledge remains unexploited for online HD map construction. For instance, OSM maps contain so-called *tags*, *i.e*. key-value pairs, such as `name: Park Avenue`, `oneway: yes`, or `lanes: 2`. The last two examples already show that the textual information is highly informative for the task of online HD map construction. In total, at the point of writing, the global OSM map contains around 100k different keys and 168M values. This vast amount of tags as well as their unstructured nature obviously cannot be exploited by handcrafted methods. Instead, we propose an NLP encoder to optimally interpret the full information contained in SD maps for the goal of HD map construction.

## 3.2   NLP Tag Embeddings

Our proposed NLP encoder takes inspiration from the area of sentence embedding in natural language processing and can generate open-vocabulary embeddings for arbitrary textual annotations. By adopting training paradigms from sentence embedding, we require no extra labels for the encoder and pre-train it in a fully self-supervised manner. Moreover, this makes the pre-training completely task-agnostic, meaning the pre-trained encoder can in principle be used in any application which desires embeddings of textual SD map annotations, not just the area of online HD map perception.

For the embedding model we choose the compact and well-studied BERT [5] architecture. SD map annotations are of considerably less complexity than the paragraphs of prose text normally used in sentence embedding, which renders large language models or other large embedding models unnecessary. A small model like BERT also enables us to run the encoder module in real time with the main online HD map perception model. The embedding dimension of the encoder is 144, with the `[CLS]` token selected as the tagset embedding, and one embedding is computed per SD map element.

**Contrastive Pretraining Objective**   Common practice in natural language processing relies heavily on pre-trained large models that are then optionally finetuned for different use cases. However the data in SD map text annotation differs strongly from the text corpus for existing pre-trained sentence embedding models. The text consists of a list of keywords rather than complete sentences and, contrary to general requirements for sentence embeddings, small changes in the text should make a big difference in the embedding. For example, a tagset with the tag `lanes: 2` should have a substantially different embedding from `lanes: 3`, even though only one letter changed.

Since a very large dataset for the use case exists as well with OpenStreetMap [22], we decided to train our embedding model from scratch. This also affords us more control of model hyperparameters such as embedding size and is necessary for reaching real-time performance. For our pre-training we adapt established methods and design a self-supervised contrastive learning objective based on the multiple negatives ranking loss [7]. Our customized objective is motivated by the fact that many annotation artifacts and semantically irrelevant tags exist in crowdsourced maps. For instance, many elements in OSM contain IDs and other cataloging information of national geodetic reference systems that are not informative for a downstream task. Our customized loss forces the model to ignore these irrelevant tags, which we identified using the main OSM tag database.

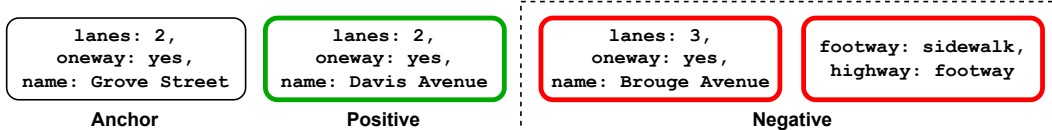

Figure 3: Example of the tag embedding contrastive pretraining objective. A positive sample is selected from tagsets with the same semantically meaningful tags, but different not meaningful ones (like the street name). Negative samples are selected from all other unique tagsets. The number of negative samples in practice is much larger than depicted here to prevent unstable training.

Figure 3 shows an example of a training batch for the customized loss. Using the dataset of all unique tagsets from the OpenStreetMap planet map, the goal of the model is to minimize the distance of two tagsets with the same semantically meaningful tags while maximizing the distance towards other tagsets. A large batch size of negative samples is needed to keep the training stable [7], 5120 in our case. We randomly sample 20 pairs of positive samples per unique relevant tagset for the training data and choose a training time of 4 epochs to ensure convergence. To facilitate further research in this area, the pre-trained encoder together with the extracted OSM annotation dataset used for training will be released alongside the main model.

### 3.3 SD Map Encoder

Following the NLP tag embedding, the SD map encoder combines these embeddings with the associated points and the scene-level context through a transformer module. The SD map encoder module is based on the architecture proposed by SMERF [21], however with key changes that significantly improve performance in the online HD map construction task and allow for much more flexibility in its input representation. A detailed visualization of the SD map encoder design with all its components can be found in Figure 4. All changes are investigated individually in Section 4.3 and Appendix A.1 and we show that their combined application significantly increases performance.

**Point Level Queries**  In SMERF [21], one query token represents one polyline, which is misaligned with the detection transformer architectures in HD map construction, that use one detection query for each point instead [15, 16, 3, 4, 30]. To align the token design, we choose one query token per point. This also has the benefit of an easier introduction of additional element types of points and relations. As all polyline elements in vectorized HD map construction architectures are detected with a fixed point number, we also resample all SD map polylines to the fixed point number of 10. Furthermore, we adopt the sin/cos positional encoding for point coordinates from SMERF [21] in our encoder and concatenate the tag embedding of the respective map element to it.

**ORF Element Identifiers**  While point-level queries better align the element representations and provide additional flexibility, they introduce a problem: Without additional changes, context for the model is lost as to which element a point belongs to, *e.g.* in the case of polyline features. We resolve this issue by taking inspiration from the field of graph transformers, where [10] show orthogonal random features (ORF) to be strong identifier features in the context of graph nodes and edges.

Following [10] and [29], we retrieve ORF features from the rows of the random orthogonal matrix $\mathbf{Q} \in \mathbb{R}^{n \times n}$, created from the QR decomposition of a random Gaussian matrix $\mathbf{G} \in \mathbb{R}^{n \times n}$. Each element is then assigned one ORF feature as its identifier, also visualized in Figure 4.

As in [10], graph nodes, or in our case, point features and polyline features, have the same ORF feature vector stacked twice in their query. Points are represented as a single query of the SD map encoder with two identical ORF identifiers, a tag embedding, and positional encoding. Polylines are represented by a list of multiple point-like queries that all share the same two identical ORF identifiers. Relations are represented using pairs of ORF identifiers, each corresponding to one of the members of a relation edge, as done traditionally in graph-structured data with node IDs. This simple graph representation method for the SD map encoder is therefore not only a identifier for map element context, but also vastly broadens the element types that can be used for prior information.

**Connection with Map Decoder**  There exist many possible ways to supply the encoded prior information to the main online HD map construction model, ranging from fusion with the BEV

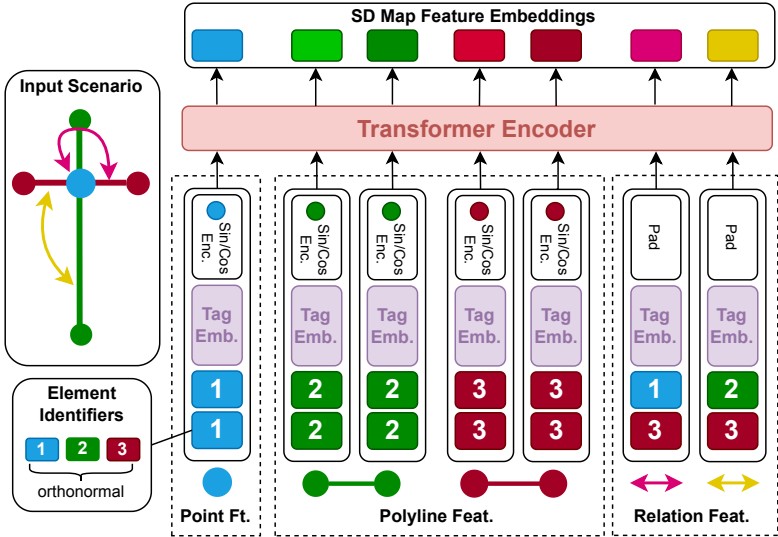

Figure 4: Detailed design of the SD map encoder and its queries. Each point query is composed of the positional sin/cos encoding of the point, the respective tag embedding and orthogonal random features (ORF) [29] that function as element identifiers and can model graph edges.

features to direct supply to the map decoder. We follow the approach of PMapNet [9], which uses an additional cross-attention layer in the map decoder as the supply modality. Compared to other approaches such as concatenation to the image BEV features, cross-attention is more expressive and can compensate spatial alignment issues better, resulting in the best performance in [9].

## 4 Experiments

We evaluate our method on the Argoverse 2 [25] and nuScenes [1] datasets, viewing Argoverse 2 as the main dataset for evaluation. We discuss our selected datasets and metric in Section 4.1 and the implementation and baselines in Section 4.2. Section 4.3 contains the performance results of SDTagNet in comparison with existing methods and ablation studies to assess the contributions of individual components in the SD map encoder. More detailed ablation studies regarding the optimization strategies applied to the NLP tag embedding module and fine-tuning training regimes can be found in Appendix A.1.

### 4.1 Datasets and Metric

To show the untainted performance of our model, we train and evaluate SDTagNet and all baselines on the geographically non-overlapping split proposed by [18]. We put a focus on the more recent Argoverse 2 dataset since it is significantly larger than nuScenes (158k vs. 30k samples) and has – to our experience – more accurate maps. As ground truth HD maps, we use recently published labels [8] that contain topologically relevant centerline paths from [17] as well as semantically more challenging solid and dashed dividers, and correct many consistency errors in the original labels [3].

Additionally, as SD maps, OpenStreetMap data corresponding to the frame-wise local HD map label is extracted from the planetary OSM database. We provide this data as SD map prior in two evaluation ranges for online HD map construction: near range ($60\,\mathrm{m} \times 30\,\mathrm{m}$) and far range ($120\,\mathrm{m} \times 60\,\mathrm{m}$), making sure to include all SD map elements that intersect that area. To the best of our knowledge, no previous method that evaluates on the increasingly important far range setting with a geographically disjoint train/val split [9, 30] combines these important evaluation settings. Comparing Table 7 of Appendix A.4 with Table 1 shows that SD priors aggravate the localization overfit observed by [18] when train and evaluation data overlaps geographically. We assume that this is due to the additional leaked information available to the network when using SD maps as prior. All evaluation is carried out with the standard mean Average Precision (mAP) metric with 0.5, 1.0 and 1.5 meters thresholds for the Chamfer distance. This matches the standard mAP metric used by our baseline architecture [16].

Table 1: Comparison of SD map prior encoding methods on Argoverse 2 [25], with the geographical split of [18]. *: With the 7 classes from [21] in the input features, which are not used in the original work. †: With OSM nodes in the input features, which are not used in the original work. All models are trained for 24 epochs.

| *Dataset: Argoverse 2* | **Near Range** *(60 m × 30 m)* | | | | | | |
|---|---|---|---|---|---|---|---|
| **Method** | $AP_{dsh}$ | $AP_{sol}$ | $AP_{bou}$ | $AP_{cen}$ | $AP_{ped}$ | **mAP** | vs. [16] |
| MapTRv2 [16] | 37.9 | 55.0 | 49.7 | 48.2 | 41.7 | 46.5 | - |
| + PMapNet [9] | 36.7 | **55.4** | 49.7 | 49.5 | 43.3 | 46.9 | +0.4 |
| + PMapNet*[9] (all info.) | **39.4** | 54.6 | 50.6 | 47.6 | 42.7 | 47.0 | +0.5 |
| + SMERF [21] | 39.4 | 54.9 | 49.4 | 49.0 | 39.0 | 46.3 | -0.2 |
| + SMERF$^†$ [21] (all info.) | 38.0 | 54.4 | 50.0 | 48.5 | 38.2 | 45.9 | -0.6 |
| + **SDTagNet** | 36.0 | 55.2 | **53.3** | **52.6** | **43.3** | **48.1** | **+1.6** |
| | **Far Range** *(120 m × 60 m)* | | | | | | |
| MapTRv2 [16] | 9.5 | 15.0 | 11.3 | 17.5 | 11.7 | 13.0 | - |
| + PMapNet [9] | 9.7 | 16.4 | 13.9 | 19.6 | 16.8 | 15.3 | +2.3 |
| + PMapNet*[9] (all info.) | 9.3 | 15.1 | 15.0 | 20.0 | 18.9 | 15.7 | +2.7 |
| + SMERF [21] | 8.7 | 14.2 | 10.9 | 15.8 | 11.5 | 12.2 | -0.8 |
| + SMERF$^†$ [21] (all info.) | 9.1 | 15.5 | 14.0 | 19.0 | 13.5 | 14.2 | +1.2 |
| + **SDTagNet** | **13.0** | **18.4** | **17.7** | **22.6** | **22.9** | **18.9** | **+5.9** |

## 4.2 Implementation Details and Baselines

Our baseline SD map encoding approaches are the two recent works PMapNet [9] and SMERF [21] that represent the two general existing approaches towards SD map encoding: Rasterization into a BEV image in the case of PMapNet [9] and encoding of the vectorized polylines via a transformer module in the case of SMERF [21]. To minimize the influence from other priors like temporal information, we select the popular single-shot architecture MapTRv2 [16] as a base model. For fair comparison, all SD map encoder modules have been ported to the MapTRv2 base architecture, thus isolating any performance differences to the SD map encoder module. All models are trained on 4 H100 Nvidia GPUs for 24 epochs on Argoverse 2 and 110 epochs for nuScenes.

Both baselines are evaluated in two settings: With their original input data settings and with all information that is available in the OSM map, to the extent that the encoding method permits its integration. For PMapNet [9], we add the more detailed road label classes from SMERF to the raster image and for SMERF [21] we introduce node features that are not used in the original work. This additionally eliminates any differences between available input data and compares SDTagNet with the fairest versions of existing SD map encoding methods. A detailed discussion of baseline implementation details and hyperparameters can be found in Appendix A.2.

## 4.3 Online HD Map Construction Performance

In this section we first discuss our results on Argoverse 2 and nuScenes in the near range and far range setting in comparison with previous work. We also present ablation studies that show the effectiveness of the proposed architectural changes and NLP tag embedding module.

**Results on Argoverse 2** Table 1 shows the results of SDTagNet on Argoverse 2 [25] in the near range and far range setting in comparison with the SD map encoding baselines PMapNet [9] and SMERF [21], applied to the base architecture MapTRv2 [16].

In the far range, all approaches using SD prior maps improve upon the prior-less MapTRv2 baseline. This shows the benefit of prior knowledge to drive safely and comfortably even at higher velocities, requiring farther perception range where camera resolution and road visibility diminish. Surpassing all other approaches, SDTagNet outperforms MapTRv2 by +5.9 mAP (+45%). Furthermore, SDTagNet beats existing SD map prior methods by +3.2 mAP (+20%), even when they are extended to use all available information. Comparing the SD map baselines, PMapNet shows better performance than SMERF, suggesting that the SMERF encoder originally used in lane topology prediction is ill-adapted to perform vectorized HD map construction.

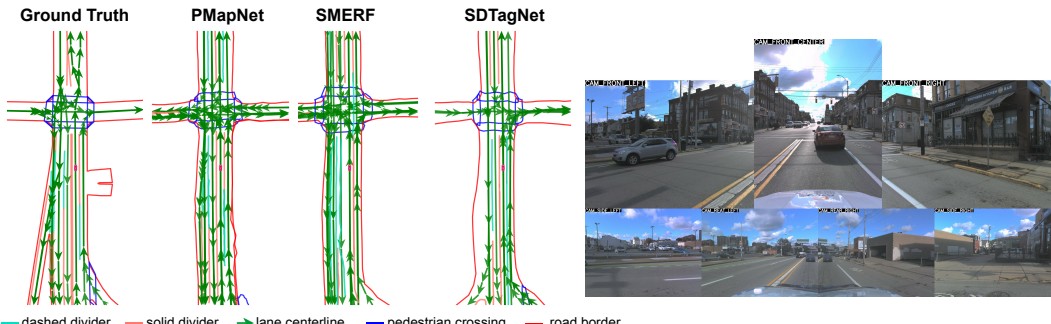

Figure 5: Qualitative comparison of SDTagNet with PMapNet (all info.) and SMERF (all info.) on Argoverse 2 in the far range setting. Both SMERF and PMapNet fail to identify the one-way road and hallucinate a standard two-way crossing topology instead. SDTagNet can translate the information in the SD map tags to a correct one-way topology prediction.

In the near range no significant performance benefit compared to the baseline can be observed for any method, even slightly reducing performance in some cases. This can be attributed to geographic overlap between train and val split for both PMapNet and SMERF as well as weaker base architectures of HDMapNet [13] and MapTR [15] in case of PMapNet. These differences emphasize the importance of using geographically non-overlapping splits, especially for models using map priors.

To compare SDTagNet with recent state-of-the-art non SD map prior methods, we also evaluated it in the more limited label setting of [3], the results of which are shown in Table 2. SDTagNet maintains competitive performance in the near range, while significantly outperforming existing approaches in the far range, acheiving a +6.1 mAP gain vs. MapTracker [3]. This is despite SDTagNet being the only single-shot method in this comparison other than HIMap [33].

The results show the strengths of SDTagNet in fully exploiting all information contained in SD prior maps, significantly outperforming state-of-the-art non SD map prior methods and existing SD map encoders, even when beneficial changes are introduced to them. A qualitative comparison of SDTagNet with PMapNet and SMERF on Argoverse 2 in the far range setting is displayed in Figure 5.

**Ablation Studies**    We conduct ablation studies on the Argoverse 2 dataset to assess the contributions of individual components in the SD map encoder, displayed in Table 3. Only the full configuration of SDTagNet, combining all proposed modifications, yields significant performance gains compared to the SMERF [21] baseline. The results further highlight the importance of ORF identifiers, without which the performance even drops when switching to point level queries. This suggests that the provided element identifiers are essential for enabling effective point-level queries.

To investigate the viability of fusion approaches for the map decoder connection as proposed by TopoSD [28], we investigated a combination of direct cross-attention to the SD map encoder tokens and cross-attention to additional SD map BEV features, shown as "+ BEV Ft." in Table 3. The SD map BEV features are rasterized similarly to PMapNet [9], but contain processed SD map encoder tokens instead of grayscale images. Despite class-level differences, this combined connection showed no overall performance gain. Our results indicate that, when evaluated on a geographic split with the more diverse labels of M3TR [8], regular cross-attention alone is sufficiently expressive on its own.

**Results on nuScenes**    Table 4 shows the evaluation results on the nuScenes dataset. The relative improvement of SDTagNet in the far range setting is even stronger, increasing performance compared to MapTRv2 by +4.1 mAP (+105%) and compared to existing methods by +2.1 mAP (+35%). This coincides with the general drop in mAP for all methods on the nuScenes geo split [18] and shows the benefits of our approach even on smaller datasets with less generalization capability. SMERF outperforms PMapNet here, indicating that these methods may be more susceptible to dataset differences than SDTagNet, which shows similar performance benefits in both cases. In the near range, the results mostly mirror those on Argoverse 2, with no method able to provide a significant performance gain. On nuScenes in particular, we observed some significant differences between the roads and structures contained in the SD map and the ground truth HD map (see Section 5). We

Table 2: Comparison of SDTagNet with recent state-of-the-art non SD map prior methods on the more limited label setting of [3], evaluated on the Argoverse 2 original split and the geographic split of [30]. * = only total mAP reported in the original paper. †: values taken from MapTracker [3].

| _Dataset: Argoverse 2_ | | **Near Range** _(60 m × 30 m)_ | | | | |
|---|---|---|---|---|---|---|
| **Method** | **Split** | **Epochs** | **$AP_{div}$** | **$AP_{bou}$** | **$AP_{ped}$** | **mAP** |
| HIMap [33] | Og. Split | **24** | 72.4 | 73.2 | 72.4 | 72.7 |
| MapUnveiler [11] | Og. Split | 30 | 74.2 | 71.9 | 72.5 | 72.9 |
| StreamMapNet† [30] | Og. Split | 72 | 74.2 | 66.1 | 70.5 | 70.3 |
| MapTracker [3] | Og. Split | 35 | 80.0 | 73.7 | 77.0 | **76.9** |
| **SDTagNet** | Og. Split | **24** | **81.7** | **76.3** | 76.1 | **78.0** |
| StreamMapNet† [30] | [30] Geo Split | 72 | 68.2 | 63.2 | 61.8 | 64.4 |
| MapTracker [3] | [30] Geo Split | 35 | **75.1** | **68.9** | **70.0** | **71.3** |
| **SDTagNet** | [30] Geo Split | **24** | 72.0 | 67.5 | 64.0 | 67.8 |
| | | **Far Range** _(100 m × 50 m)_ from [30] | | | | |
| MapUnveiler [11] | Og. Split | 30 | 67.9 | 62.6 | 71.7 | 67.4 |
| StreamMapNet† [30] | Og. Split | 30 | -* | -* | -* | 57.7 |
| **SDTagNet** | Og. Split | **24** | **80.2** | **72.9** | **81.7** | **78.3** |
| StreamMapNet† [30] | [30] Geo Split | 72 | 56.1 | 47.5 | 60.1 | 54.6 |
| MapTracker [3] | [30] Geo Split | 35 | 64.6 | 58.5 | 71.2 | 64.8 |
| **SDTagNet** | [30] Geo Split | **24** | **69.5** | **68.0** | **75.1** | **70.9** |

Table 3: Ablation study of different encoder components on the Argoverse 2 dataset. All experiments are in the near range setting and all models are trained for 24 epochs. †: With OSM nodes in the input features, which are not used in the original work. + BEV Ft.: With BEV features as an additional prior mode similar to [28].

| _Base Enc.: SMERF† [21] (all info.)_ | | | | | **AP** | | | | |
|---|---|---|---|---|---|---|---|---|---|
| Pt. Lvl. | NLP Emb. | ORF Id. | + BEV Ft. | **mAP** | dsh. | sol. | bou. | cen. | ped. |
| - | - | - | - | 45.9 | 38.0 | 54.4 | 50.0 | 48.5 | 38.2 |
| ✓ | - | - | - | 44.3 | 34.9 | 53.0 | 49.5 | 45.5 | 38.7 |
| - | ✓ | - | - | 45.8 | 37.0 | 55.4 | 47.9 | 49.2 | 39.3 |
| ✓ | - | ✓ | - | 46.3 | 39.3 | 53.3 | 49.6 | 48.5 | 40.8 |
| ✓ | ✓ | - | - | 45.0 | 33.6 | 53.2 | 50.2 | 47.7 | 40.6 |
| ✓ | ✓ | ✓ | - | **48.1** | 36.0 | 55.2 | 53.3 | 52.6 | 43.3 |
| ✓ | ✓ | ✓ | ✓ | **48.1** | 38.2 | 56.3 | 52.6 | 50.7 | 42.3 |

hypothesize this is one of the main causes of the lacking performance in the near range, where the visibility of road features is much higher and the model is less reliant on SD map features.

## 5   Limitations

Despite SDTagNet's strong performance, we observe several limitations. First, like all SD map prior methods, SDTagNet assumes congruency of SD map and HD map ground truth. Depending on the level of detail and quality of dataset ground truth and SD map, a number of map discrepancies can be observed. This is especially the case for crowdsourced databases like OSM, as shown for Argoverse 2 by [20] and nuScenes by [9]. We also noted significant differences on nuScenes, particularly concerning structures and roads that are missing from the ground truth HD map. Figure 6b shows examples of a tunnel and smaller service roads that are not included in the nuScenes map, impacting model accuracy and training data quality.

Another limitation concerns the case of one-way roads. SDTagNet is able to correctly identify one-way roads from the SD map tag information, however in some cases the driving direction is predicted wrongly. An example of this is displayed in Figure 6a. Since OSM encodes one-way direction via polyline point order, this likely stems from the lack of point order information in the element identifiers. Adding point order identifiers for polyline features could help remedy this limitation.

Table 4: Comparison of SD map prior encoding methods on nuScenes [1], with the geographical split of [18]. *: With the 7 classes from [21] in the input features, which are not used in the original work. †: With OSM nodes in the input features, which are not used in the original work. All models are trained for 110 epochs.

| *Dataset: nuScenes* | **Near Range** *(60 m × 30 m)* | | | | | | |
|---|---|---|---|---|---|---|---|
| **Method** | **AP$_{dsh}$** | **AP$_{sol}$** | **AP$_{bou}$** | **AP$_{cen}$** | **AP$_{ped}$** | **mAP** | vs. [16] |
| MapTRv2 [16] | 12.5 | 19.1 | 32.4 | 29.1 | 21.6 | 22.9 | - |
| + PMapNet [9] | 13.4 | 20.9 | 32.5 | 30.2 | **22.2** | **23.9** | +1.0 |
| + PMapNet*[9] (all info.) | 12.6 | 19.8 | 32.6 | 28.7 | 21.1 | 23.0 | +0.1 |
| + SMERF [21] | 14.6 | 19.3 | **34.2** | 28.9 | 21.6 | 23.7 | +0.8 |
| + SMERF† [21] (all info.) | 15.5 | **20.2** | 33.0 | 28.4 | 19.8 | 23.4 | +0.5 |
| **+ SDTagNet** | **19.3** | 19.8 | 31.3 | **30.4** | 15.6 | 23.3 | +0.4 |
| | **Far Range** *(120 m × 60 m)* | | | | | | |
| MapTRv2 [16] | 2.7 | 3.7 | 4.6 | 5.4 | 2.9 | 3.9 | - |
| + PMapNet [9] | 2.4 | 3.5 | 5.0 | 5.8 | 2.9 | 3.9 | +0.0 |
| + PMapNet*[9] (all info.) | 2.6 | 3.6 | 4.7 | 5.6 | 2.7 | 3.8 | -0.1 |
| + SMERF [21] | 4.5 | 5.6 | 8.1 | 7.6 | 3.1 | 5.8 | +1.9 |
| + SMERF† [21] (all info.) | 4.1 | **6.6** | 7.8 | 7.5 | 3.4 | 5.9 | +2.0 |
| **+ SDTagNet** | **8.3** | **6.6** | **10.3** | **9.3** | **5.3** | **8.0** | **+4.1** |

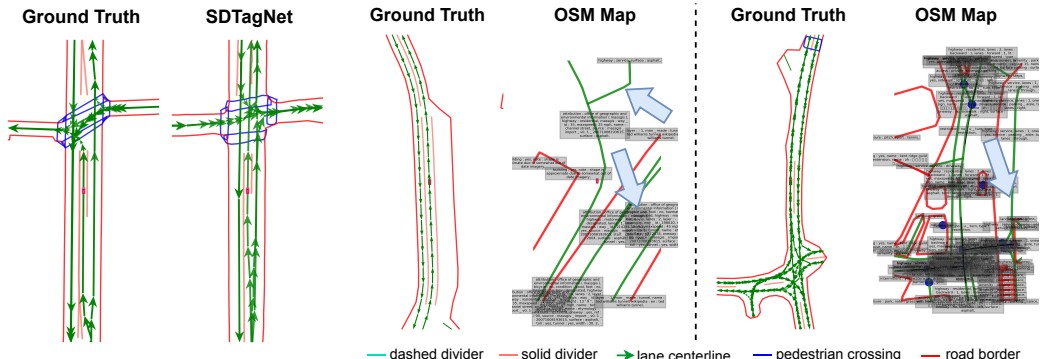

— dashed divider — solid divider → lane centerline — pedestrian crossing — road border

(a) Pred. false one-way direction.  (b) Discrepancies between the SD map and ground truth.

Figure 6: Examples for observed limitations of SDTagNet in Argoverse 2 (left) and nuScenes (right). Figure 6a: Even when correctly recognizing a one-way road, SDTagNet can fail to predict its correct driving direction. Figure 6b: Discrepancies between dataset HD maps and OSM SD maps like missing tunnels (left) or small service roads (left, right).

# 6  Conclusion

We present SDTagNet, the first online HD map construction approach that fully utilizes commonly available SD maps, including all elements such as points, polylines and relations in conjunction to their descriptive textual information. This is achieved by incorporating embeddings of the textual annotations for each point, polyline, and relation in addition to their geometric information, encoding their key-value tags with a contrastively pre-trained and fine-tuned BERT encoder. For scene-level encoding of SD map priors with their geometric data we increase the query resolution from polyline-level to point-level using orthogonal element identifiers, allowing for greater expressiveness by integration of point and relational SD map elements in addition to the established polylines.

The combination of these encoding methods results in a 20-35% increase in performance on Argoverse 2 and nuScenes, respectively, when evaluated on a far perception range. This makes SDTagNet the first SD map prior based online HD map construction model free from manually selected features, with scalable self-supervised pre-training on planet-scale SD map data, improving the performance of existing methods by large margins.

# 7   Acknowledgements

The authors gratefully acknowledge the computing time provided on the high-performance computer HoreKa by the National High-Performance Computing Center at KIT (NHR@KIT). This center is jointly supported by the Federal Ministry of Education and Research and the Ministry of Science, Research and the Arts of Baden-Württemberg, as part of the National High-Performance Computing (NHR) joint funding program (https://www.nhr-verein.de/en/our-partners). HoreKa is partly funded by the German Research Foundation (DFG). SD map data copyrighted OpenStreetMap contributors and available from https://www.openstreetmap.org.

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

Table 5: Ablation studies concerning the NLP tag embedding module on the Argoverse 2 dataset. All experiments are in the near range setting and all models are trained for 24 epochs. rel. tag CL = relevant tag contrastive learning, meaning whether our contrastive learning objective based on relevant tags is used during pretraining or not. If not, irrelevant tags are removed in preprocessing and the positive sample is also the anchor for pretraining.

| Fine-tuning LR | Rel. Tag CL | mAP | AP | | | | |
| --- | --- | --- | --- | --- | --- | --- | --- |
| | | | dsh. | sol. | bou. | cen. | ped. |
| Frozen | ✓ | 45.6 | 36.5 | 54.2 | 49.8 | 48.4 | 39.2 |
| × 0.5 | ✓ | 47.1 | 35.7 | 53.5 | 52.0 | 50.0 | 44.3 |
| × 0.1 | - | 47.5 | 35.2 | 54.5 | 51.9 | 50.9 | 45.0 |
| × 0.1 | ✓ | **48.1** | 36.0 | 55.2 | 53.3 | 52.6 | 43.3 |

Table 6: Ablation studies of the SD map encoder cross attention on the Argoverse 2 dataset. All experiments are in the near range setting and all models are trained for 24 epochs.

| Att. Heads | Dropout | mAP | AP | | | | |
| --- | --- | --- | --- | --- | --- | --- | --- |
| | | | dsh. | sol. | bou. | cen. | ped. |
| 8 | 0.2 | 46.9 | 35.7 | 53.7 | 50.6 | 52.0 | 44.3 |
| 4 | 0.1 | 47.7 | 38.6 | 54.3 | 51.9 | 51.4 | 42.3 |
| 8 | 0.1 | **48.1** | 36.0 | 55.2 | 53.3 | 52.6 | 43.3 |

# A    Technical Appendices and Supplementary Material

The supplementary material focuses on three key aspects: Additional ablation studies on all individual components of SDTagNet (appendix A.1) and additional experiments on a non-geo split (appendix A.4), implementation details of the baselines and NLP pretraining (appendix A.2 and appendix A.3) as well as more qualitative examples and visualizations (appendix A.7).

## A.1    Ablation studies

Additional ablation studies concerning the optimization strategies applied to the NLP tag embedding module and fine-tuning training regimes, conducted on the Argoverse 2 dataset, are summarized in Table 5, and Table 6.

For the NLP tag embedding module, we observe in Table 5 that fine-tuning the pretrained language model with a small learning rate is crucial to align the self-supervised embeddings with the specific objectives of the downstream task. Additionally, removing our custom contrastive pretraining objective results in degraded performance. This confirms the utility of our training scheme, which guides the model to focus on semantically meaningful tag representations.

Additionally, we tested different numbers of attention heads and different values of dropout for the SD map encoder cross-attention in Table 6, settling on 8 attention heads and a dropout of 0.1 as the configuration with the best performance.

## A.2    Implementation Details of Baseline Approaches

We reproduce our baselines PMapNet [9] and SMERF [21] according to their publicly available source code, with a few changes to adapt to our different experiment setting and base architecture.

For the PMapNet base setting, we create a BEV grid with the same resolution as the main image BEV features and rasterize point and polyline features on this grid. Polyline and point features are drawn with a fixed width of 7 an 5 grid cells, respectively. Like in the PMapNet reference implementation, each element gets assigned a descending grayscale value from 0 to 255, without any additional labels. For the enhanced version PMapNet (all info.), we additionally add the 7 classes defined in SMERF, plus an additional `other` class for elements that do not belong to the predefined classes. Follwing the

Table 7: Comparison of SD map prior encoding methods on the Argoverse 2 data set, with the original geographically overlapping split. †: With OSM nodes and relations in the input features, which are not used in the original work. All models are trained for 24 epochs.

| *Dataset: Argoverse 2, og. split* | **Near Range** *(60 m × 30 m)* | | | | | | |
|---|---|---|---|---|---|---|---|
| **Method** | $AP_{dsh}$ | $AP_{sol}$ | $AP_{bou}$ | $AP_{cen}$ | $AP_{ped}$ | **mAP** | vs. [16] |
| MapTRv2 [16] | **61.1** | 66.8 | 67.9 | 61.9 | 58.6 | 63.3 | - |
| + SMERF† [21] (all info.) | 60.6 | 66.6 | 67.8 | 60.1 | 60.5 | 63.1 | -0.2 |
| **+ SDTagNet** | 59.5 | **67.7** | **70.7** | **65.5** | **62.1** | **65.1** | **+1.8** |
| | **Far Range** *(120 m × 60 m)* | | | | | | |
| MapTRv2 [16] | 26.7 | 33.4 | 22.9 | 30.7 | 30.6 | 28.9 | - |
| + SMERF† [21] (all info.) | 30.7 | 37.9 | 28.9 | 33.9 | 35.3 | 33.4 | +4.5 |
| **+ SDTagNet** | **39.6** | **45.1** | **38.7** | **43.8** | **45.5** | **42.5** | **+13.6** |

original source, the feature grids of both PMapNet versions get processed by a simple 3-layer CNN before being supplied to the main map decoder.

For the SMERF base setting, we keep all hyperparameters as found in the original source code, except for also including the additional `other` class for elements that do not belong to the predefined classes. We also select a fixed polyline point number of 10 like in the other experiments instead of 11 for easier preprocessing. For the enhanced version SMERF (all info.), we add point features to the encoder which also get assigned the `other` class. To fit the point features into the original polyline-level query token, we repeat the single point for 10 times. We do not include relation features in both PMapNet and SMERF, as the encoder architectures make this not possible without major changes.

## A.3   Implementation Details Contrastive Pretraining Objective

In this section, we describe in more detail the selection process for the non-relevant tags used in the contrastive pretraining objective. The selection was performed with the help of the central OSM tag database located at `https://taginfo.openstreetmap.org/` and for all tags that were used at least 100,000 times. For comparison, as of the time of writing, the most used tag key is `building` with around 650 million uses, or around 6% of total objects. As a rule, the selection was performed task-agnostically and did not consider whether a tag was particularly relevant for the task of online HD map construction. This preserves the general applicability of the proposed embedding model. Two main categories of tags are considered not informative. First, tags that are names and other identifying information of places like telephone numbers or website links. Second, map annotation artifacts, primarily IDs and other data from various government geodetic databases that were the original source of the data. As an example, most OSM elements in the United States contain IDs from the Topologically Integrated Geographic Encoding and Referencing system (TIGER) by the US Census Bureau, from which these elements were imported. The full list contains around 110 tag keys in total and can be found in the accompanying source code.

## A.4   Argoverse 2 Original Split

To examine the differences to a non-geographic split setting, we furthermore evaluated a reduced model set of MapTRv2, SMERF (all info.), and SDTagNet on the original Argoverse 2 split with about 40% geographic overlap [18]. The results in Table 7 show that SDTagNet maintains superior performance in both range settings, with a much higher absolute mAP gain in the far range of +9.1 mAP compared to SMERF. It worth noting that the base performance is already 15 to 20 mAP higher in both settings, confirming the findings in [18] and indicating that significant geographic overfitting still takes place in the training of online HD map construction models.

## A.5   SD Map Augmentation Methods

Motivated by the findings above and a general trend of increased overfitting we observed with SD map priors, we explored augmentation methods for SD map priors that could increase the performance

Table 8: Comparison of different SD map prior augmentation methods on the Argoverse 2 dataset. All experiments are in the near range setting and all models are trained for 24 epochs. Loc. Const. = locally constant, meaning all sd map elements in one frame are augmented with the same offset and rotation.

| | Pos. Noise | | | | AP | | | | |
| El. Drop Rate | Loc. Const. | $\sigma_{\text{trans}}$ | $\sigma_{\text{rot}}$ | mAP | dsh. | sol. | bou. | cen. | ped. |
|---|---|---|---|---|---|---|---|---|---|
| 0.1 | - | 1 m | 2° | 43.8 | 40.2 | 54.6 | 47.3 | 43.6 | 33.3 |
| 0.1 | ✓ | 1 m | 2° | 45.6 | 37.3 | 54.5 | 48.8 | 47.4 | 39.9 |
| - | - | - | - | **48.1** | 36.0 | 55.2 | 53.3 | 52.6 | 43.3 |

Table 9: Comparison of different SD map prior tag masking augmentation methods on the Argoverse 2 dataset. All experiments are in the near range setting and all models are trained for 24 epochs. El. Aug. Rt. = Rate of elements where tag masking is applied. Non-Rel. Only = Tag masking is only applied to non-relevant tags. NLP Ft. LR = Fine-tuning learning rate multiplier for the NLP encoder.

| | Tag Masking | | | | AP | | | | |
| El. Aug. Rt. | Tag Drop Rt. | Non-Rel. Only | NLP Ft. LR | mAP | dsh. | sol. | bou. | cen. | ped. |
|---|---|---|---|---|---|---|---|---|---|
| 0.5 | 0.4 | - | × 0.1 | 44.7 | 37.4 | 54.2 | 47.5 | 46.7 | 37.9 |
| 0.5 | 0.6 | ✓ | × 0.1 | 47.2 | 36.2 | 54.6 | 51.6 | 49.2 | 44.7 |
| 0.5 | 0.6 | ✓ | × 0.5 | 47.4 | 34.7 | 55.3 | 52.1 | 51.5 | 43.2 |
| - | - | - | × 0.1 | **48.1** | 36.0 | 55.2 | 53.3 | 52.6 | 43.3 |

outside of the geographic training area. [28] showed that position noise augmentation for the SD map prior input during training increases performance when noisy priors are also there during validation. In this section however, our goal is to examine whether SD map prior augmentation can increase the *general* performance of a model, similar to the common use of augmentation techniques in computer vision. Additionally, with our direct use of the text annotations, we have an additional avenue for augmentation available in modifying the text content of these annotations.

We focused on two main augmentation modes: Position noise and element dropping, shown in Table 8, and text annotation tag masking, shown in table 9. Unfortunately, in both cases, we were unable to note a performance benefit when evaluated on the validation set without any added noise. For the element-level noise modes in Table 8, both augmentation variants worsened performance instead, even when all SD map elements in one frame were moved with the same offset to simulate localization noise. The tag masking augmentation in table 9 uses an element augmentation rate to randomly select what percentage of elements will be modified and a tag drop rate to select how many tags will be dropped. An additional parameter considered whether only non-relevant tags are dropped or not. The performance of this augmentation mode is higher than with position noise, especially when only non-relevant tags are dropped. Nonetheless, tag masking was also not able to improve performance compared to the baseline without any augmentation. We believe that a method to prevent geographic overfitting via augmentation techniques or in other ways would significantly increase generalization and overall performance in online HD map construction, particularly when priors are also available.

### A.6 Inference Time and Model Size

SDTagNet is designed with real-time capability in mind: the NLP tag encoder is based on a compact BERT model (embedding dim 144), and the SD map encoder is lightweight compared to the main perception backbone. To evaluate the real time capability of SDTagNet, we have investigated its inference time in comparison with no prior and with other SD map encoders. The parameter count and FPS of SDTagNet and baselines on an NVIDIA H100 GPU can be found in Table 10. The additional inference time for SDTagNet is less than 7 ms per frame, and the memory overhead is marginal relative to the base model. We thus consider our approach suitable for real-time deployment in autonomous driving systems, with a negligible difference to other SD map encoding methods.

Table 10: Comparison of model parameter count, FPS, and VRAM with MapTRv2 and two SD map baseline methods. All experiments are on the Argoverse 2 dataset with the far-range setting, averaged over 10 samples and tested on a NVIDIA H100 GPU.

| Model | Param. Count | FPS | VRAM [MB] |
|---|---|---|---|
| MapTRv2 [16] (no prior) | 42.12 M | 16.0 | 3654 |
| MapTRv2 + PMapNet [9] | 50.48 M | 13.9 | 3698 |
| MapTRv2 + SMERF [21] | 44.62 M | 14.9 | 3662 |
| MapTRv2 + SDTagNet | 50.54 M | 14.5 | 3686 |

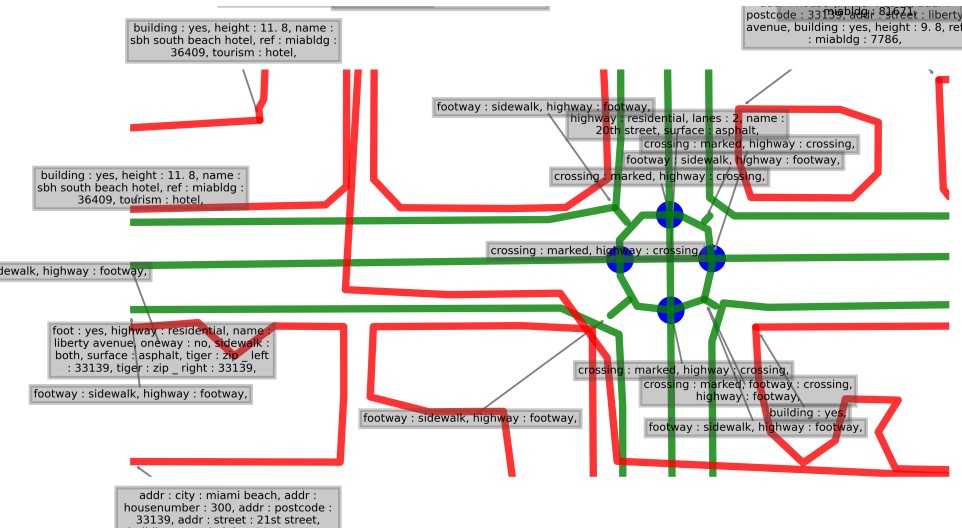

Figure 7: Larger resolution example of the OSM SD map input data on Argoverse 2 used for SDTagNet. All text annotations are used as-is, without any preprocessing or filtering. This makes the design of SDTagNet highly scalable and adaptive.

## A.7  Additional Qualitative Examples and Visualizations

**Example of OSM Text Annotations**    To provide a better understanding of the format and content of text annotations in OpenStreetMap, we include a larger resolution example of the OSM elements for a frame in fig. 7. Every element has annotated information, this includes buildings and roads with their names, oneway information or number of lanes for roads, pedestrian ways, pedestrian crossing points, etc. The road annotations on the lower left also contain some of the aforementioned TIGER geodetic database identifiers.

**Additional Examples of nuScenes Map Discrepancies**    Figure 8 shows additional examples of HD and SD map discrepancies we observed in nuScenes. These consist of inconsistent road borders that include loading and parking areas, differences in annotated crossing topology and annotation ranges that are too short and leave parts of the road unannotated. We hypothesize that these differences play a role in the reduced performance of all SD map prior methods in the nuScenes near range setting, where information from the camera images is more plentiful.

**Additional Qualitative Comparisons of SDTagNet**    In fig. 9 and fig. 10 we present additional comparisons of SDTagNet with the baselines PMapNet [9] and SMERF [21] on Argoverse 2 and nuScenes, respectively. Figure 9 contains examples that show how SDTagNet can utilize text-annotated information such as oneway roads or the number of lanes and transfer this into improved prediction results. Figure 10 displays the increased accuracy and topological consistency of SDTagNet on nuScenes compared to previous methods.

**Ground Truth**       **SD Map**

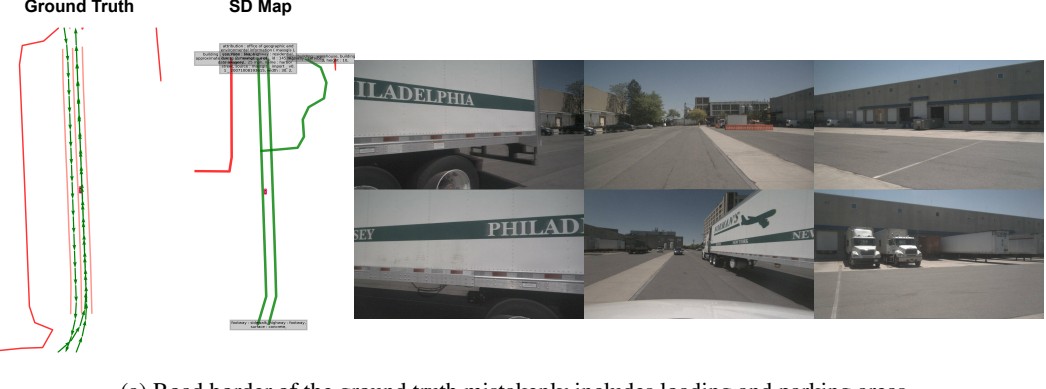

(a) Road border of the ground truth mistakenly includes loading and parking areas.

**Ground Truth**       **SD Map**

(b) Non-existent crossing arm included in the ground truth.

**Ground Truth**       **SD Map**

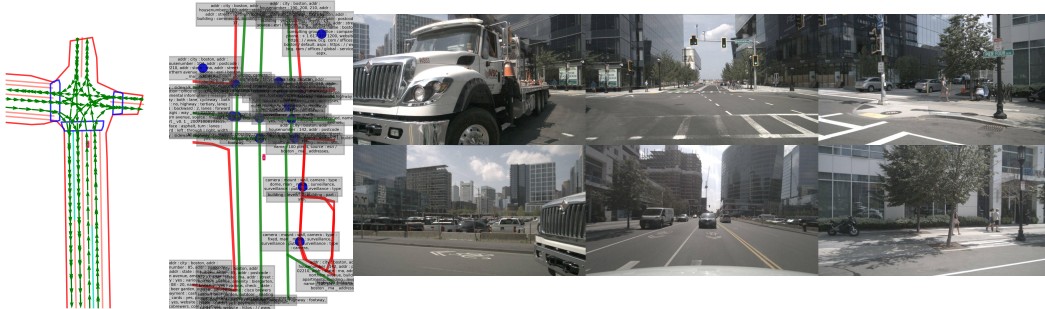

(c) Ground truth annotation cutoff is too short.

Figure 8: Additional examples of discrepancies between the ground truth HD maps and OSM SD maps in the far range setting. Inconsistent road borders, wrong crossing topologies and too short of an annotation cutoff cause inconsistencies.

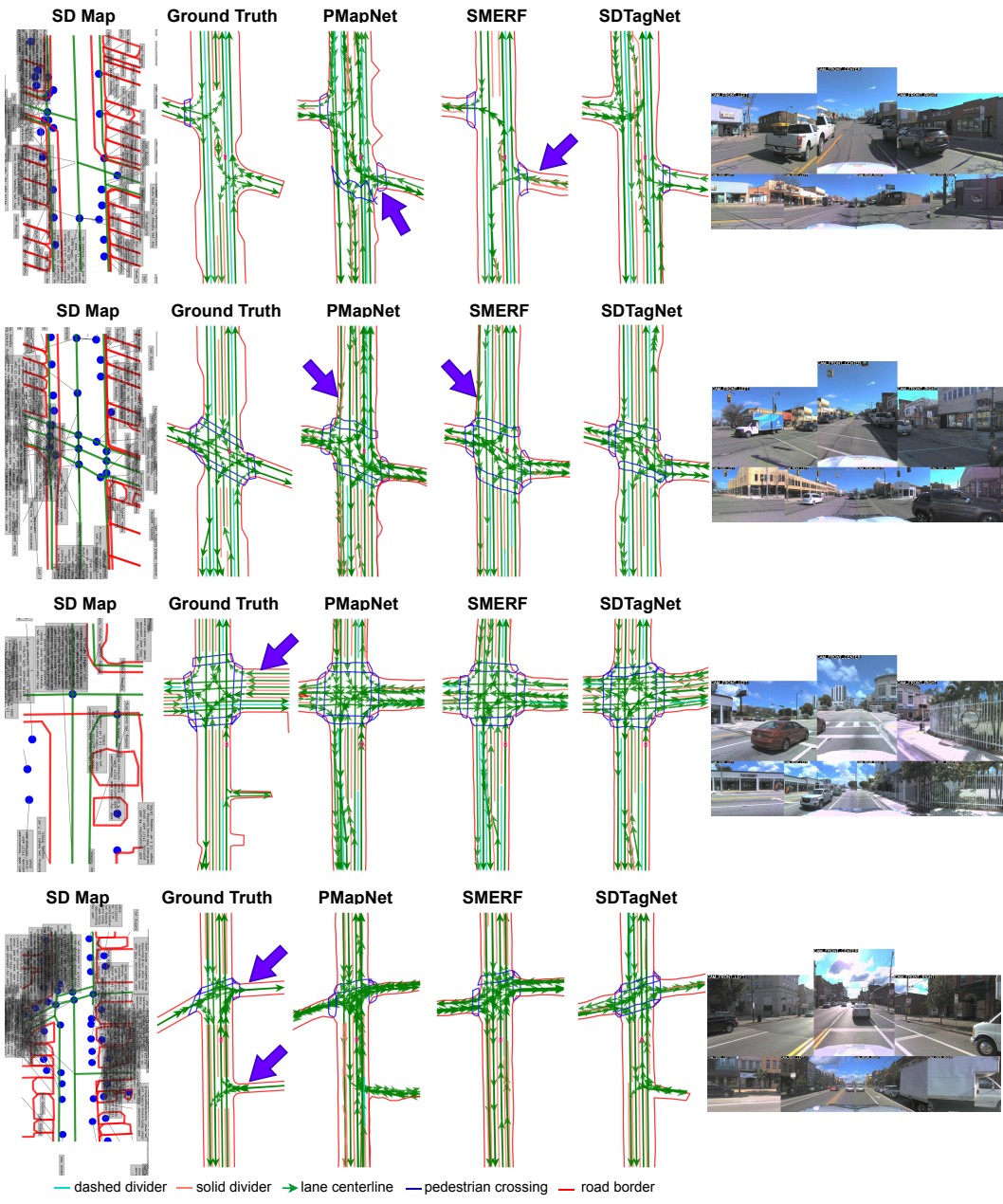

Figure 9: Further qualitative comparison of SDTagNet with PMapNet (all info.) and SMERF (all info.) on Argoverse 2 in the far range setting. SDTagNet is able to utilize text-annotated information such as number of lanes and oneway roads to improve prediction results.

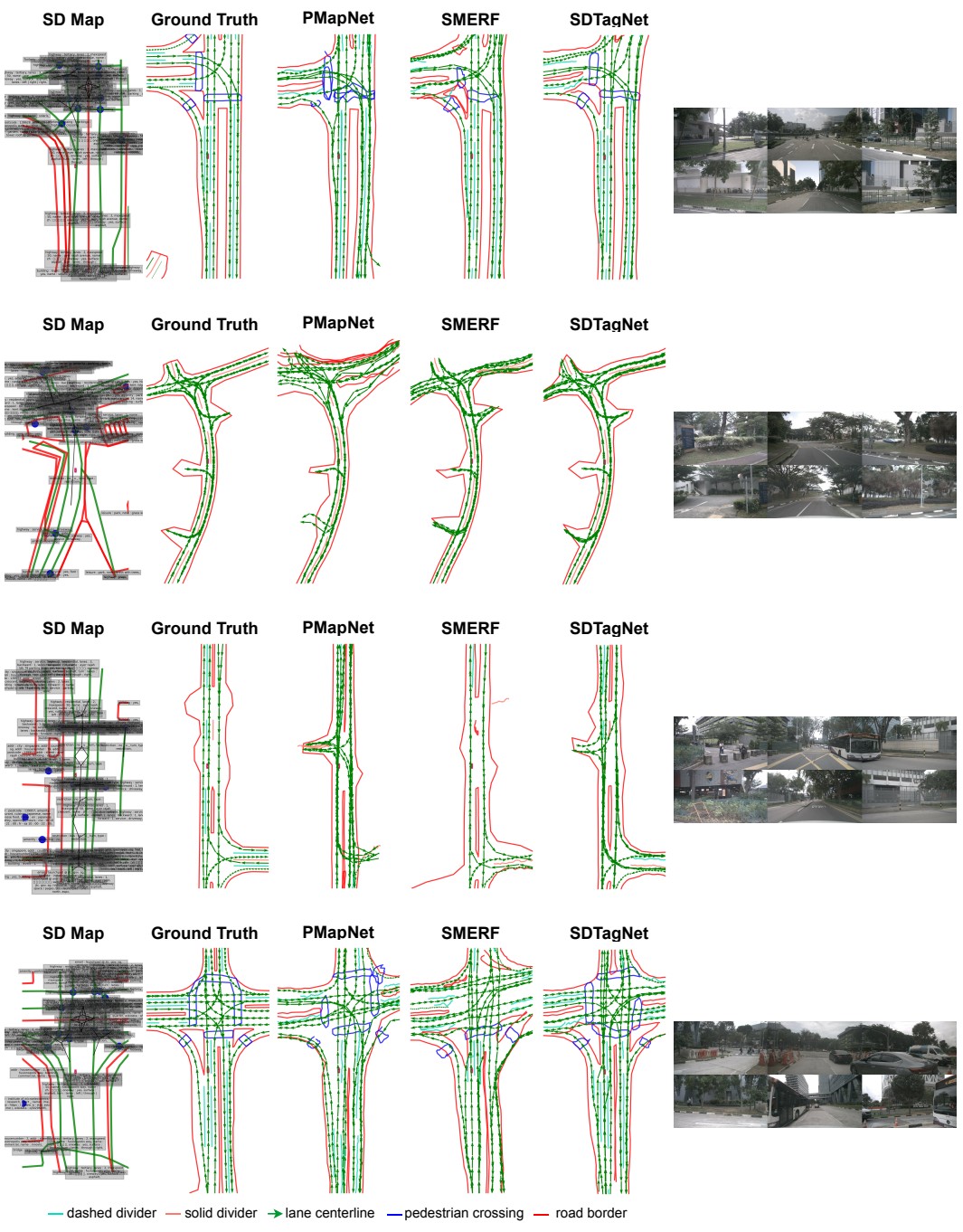

Figure 10: Qualitative comparison of SDTagNet with PMapNet (all info.) and SMERF (all info.) on nuScenes in the far range setting. SDTagNet shows less errors and more consistent topology compared to previous methods.

