# OpenReview forum: "SDTagNet: Leveraging Text-Annotated Navigation Maps for Online HD Map Construction"
_NeurIPS.cc/2025/Conference — NeurIPS 2025 poster_

### Official Review · Reviewer_CvWw · 2025-06-17

**Clarity:** 3
**Significance:** 2
**Originality:** 2
**Rating:** 4
**Confidence:** 4

**Summary:**

The paper proposes SDTagNet, an innovative method for online HD map construction, which leverages text-annotated SD maps (e.g., OpenStreetMap) as priors to improve the performance of high-definition (HD) map generation. The method integrates semantic information from textual annotations with geometric features to enhance long-range detection accuracy. Experiments on two prominent datasets (Argoverse 2 and nuScenes) demonstrate significant improvements in map perception accuracy.

**Questions:**

1. Given the complexity of incorporating textual annotations into the SD maps and the need for real-time map construction in autonomous driving systems, what is the expected computational cost (e.g., inference time, memory usage) of SDTagNet in a real-time setting?
2. The method relies heavily on SD map priors (e.g., OpenStreetMap) to construct HD maps. How does the method handle scenarios where SD maps are incomplete, outdated, or contain errors (such as missing or incorrect features)?
3. The evaluation is conducted on datasets like Argoverse 2 and nuScenes, but these datasets may not fully represent the diversity of geographic and infrastructural conditions that SDTagNet would encounter in real-world deployment. How well does the method generalize to different geographic regions with varying levels of map detail and urbanization?

**Ethical Concerns:**

["NO or VERY MINOR ethics concerns only"]

**Final Justification:**

The author address my concerns, so I decide to raise the rating.

**Limitations:**

Yes

**Quality:**

2

**Strengths And Weaknesses:**

Pros:
1. The integration of NLP-derived features into SD map encoding is a novel approach that expands the utility of publicly available SD maps, particularly OpenStreetMap, for autonomous vehicle navigation and online HD map construction.
2. The proposed SDTagNet architecture achieves substantial improvements over previous methods, particularly in the far range, demonstrating its effectiveness in real-world autonomous driving applications.
3. The experiments cover two well-established datasets, Argoverse 2 and nuScenes, providing a robust evaluation of the proposed method in different contexts. This strengthens the generalizability of the approach.
4. The adoption of contrastive pretraining for NLP tag embeddings allows SDTagNet to leverage a vast amount of unannotated map data, reducing the dependency on handcrafted feature engineering and improving scalability.

Cons:
1. While the paper introduces an interesting approach, the novelty of the work may be viewed as incremental rather than groundbreaking. The idea of using prior map data for HD map construction is not new, and other works have already explored using SD maps, though without the same emphasis on textual annotations.
2. Although SDTagNet excels in far-range detection, it shows limited improvement in near-range scenarios. This could be seen as a limitation if the method is to be applied in real-world driving conditions where near-range perception is equally important.
3. The method heavily relies on the quality and comprehensiveness of the SD map priors, such as OpenStreetMap. However, these maps are not always fully accurate or complete, especially in certain geographic areas where data might be sparse. The paper does not address how the method handles situations where SD map priors are missing critical details or contain erroneous information.
4. The paper does not explore the computational complexity of the model, nor does it discuss the feasibility of deploying SDTagNet in real-time systems with high computational constraints. The inference speed and memory requirements, especially with large-scale map data, are important factors for practical deployment, and they need to be addressed more thoroughly.
5. While the introduction of the point-level SD map encoder is a key innovation, the methodology lacks detailed explanation on how it processes complex map elements (points, polylines, relations) and how it handles the integration of these diverse map features at a computational level.

---

> ### Author Rebuttal · Authors · 2025-07-30
>
> # Author Response to Reviewer CvWw
>
>
> ### W1: Incremental Novelty of Using SD Maps and Textual Annotations
>
> >While the paper introduces an interesting approach, the novelty of the work may be viewed as incremental rather than groundbreaking. The idea of using prior map data for HD map construction is not new, and other works have already explored using SD maps, though without the same emphasis on textual annotations.
>
> We fully agree that SDTagNet is not the first paper to use SD maps as prior information and indeed use other SD map approaches as baselines. However, previous methods (e.g., PMapNet, SMERF) are limited to polylines with a small, hand-selected set of attributes or classes. This disregards the majority of available SD map data, including topological hints or textual information.
>
> The groundbreaking idea of SDTagNet is to fully leverage the rich, open-vocabulary textual annotations available in SD maps like OpenStreetMap, i.e. combining online HD vector map construction and natural language processing techniques for the first time.
> SDTagNet introduces a contrastively pre-trained NLP tag embedding module, enabling the use of all textual annotations *without* manual feature engineering.
>
> Our experimental results show that adding an NLP encoder is not only a substantial conceptual step forward in utilizing the full potential of SD maps. It also significantly improves performance especially for far-range detections. We therefore believe that SDTagNet represents a considerable advance beyond prior art.
>
> ### W2: Limited Improvement in Near-Range Scenarios
>
> >Although SDTagNet excels in far-range detection, it shows limited improvement in near-range scenarios. This could be seen as a limitation if the method is to be applied in real-world driving conditions where near-range perception is equally important.
>
>    We agree that the improvement in near-range perception is limited. As noted by reviewer kgfW, this is expected as near-range regions are covered well by onboard sensors, making the additional value of SD map priors naturally less pronounced.
> However SDTagNet is still able to provide a substantial benefit for classes like road boundaries or lane centerlines. We have included a more detailed performance discussion in the response to Reviewer kgfW.
>
> Hence our main focus is on still insufficient far-range perception, where SDTagNet demonstrates substantial gains (+45% mAP over no prior, +20% over previous SD map prior methods). This is particularly relevant for real-world autonomous driving, where long-range perception of HD maps is critical for safe and comfortable planning, especially at higher speeds.
>
> ### W3: Reliance on SD Map Quality and Handling Incomplete/Erroneous Priors
>
> >The method heavily relies on the quality and comprehensiveness of the SD map priors, such as OpenStreetMap. However, these maps are not always fully accurate or complete, especially in certain geographic areas where data might be sparse. The paper does not address how the method handles situations where SD map priors are missing critical details or contain erroneous information.
>
> We acknowledge that SDTagNet, like all methods using map priors, depends on the quality and completeness of the prior (SD) map.
> We have already investigated missing and noisy SD maps as augmentation techniques in Table 7 and 8 in the Appendix. However, we can use the same techniques, which cover normally distributed point noise, element and textual attribute dropout, to further analyze SDTagNet's performance in real-world scenarios with noisy or incomplete SD map data.
>
> To that end, we evaluated the original version of SDTagNet (trained without noisy SD map data) on SD maps with positional noise and/or dropout. The results are shown in the Tables 5 and 6 below. One can observe that SDTagNet generally maintains its strong performance with noisy data, even when not explicitly trained on it.
>
> #### Table 5: Influence of Noisy SD Maps
>
> | Noise in train | El. Drop Rate | Loc. Const. | σ_trans | σ_rot | **mAP** (perfect SD map prior in validation) | **mAP** (**with** noise added to SD map prior in validation ) |
> |---------------|---------------|-------------|---------|-------|---------|-------|
> |       ✓       | 0.1           | -           | 1 m     | 2°    |  **43.8**   | **43.8**   |
> |       ✓       | 0.1           | ✓           | 1 m     | 2°    |  **45.6**  | **45.6** |
> |       x       | 0.1           | -           | 1 m     | 2°    | **48.1**   |  **46.9**   |
> |       x       | 0.1           | ✓           | 1 m     | 2°    | **48.1**   |  **47.6**   |
>
> *Investigation of effects of SD maps with positional noise when used as augmentation during training (rows 1-2) as well as only at test time (rows 3-4). All experiments are on the Argoverse 2 dataset in the near range setting and all models are trained for 24 epochs. Loc. Const. = locally constant, meaning all SD map elements in one frame are distorted by the same offset and rotation*
>
> ---
>
> #### Table 6: Influence of Incomplete SD Maps
>
> | Noise in train | El. Aug. Rt. | Tag Drop Rt. | Non-Rel. Only | NLP Ft. LR | **mAP** (perfect SD map prior in validation) | **mAP** (**with** element dropout added to SD map prior in validation) |
> |---------------|-------------|-------------|---------------|------------|---------|-------|
> |       ✓       | 0.5         | 0.4         | -             | × 0.1      | **44.7**    | **44.7** |
> |       ✓       | 0.5         | 0.6         | ✓             | × 0.1      | **47.2**    | **47.2** |
> |       ✓       | 0.5         | 0.6         | ✓             | × 0.5      | **47.4**    | **47.4** |
> |       x       | 0.5         | 0.4         | -             | -      | **48.1**    | **47.1** |
> |       x       | 0.5         | 0.6         | ✓             | -      | **48.1**    | **47.7** |
>
> *Investigation of effects of incomplete SD maps when used as augmentation during training (rows 1-3) as well as only at test time (rows 4-5). All experiments are on the Argoverse 2 dataset in the near range setting and all models are trained for 24 epochs. El. Aug. Rt. = Rate of elements where tag dropout is applied. Non-Rel. Only = Tag dropout is only applied to non-relevant tags. NLP Ft. LR = Fine-tuning learning rate multiplier for the NLP encoder.*
>
> ### W4: Computational Complexity and Real-Time Feasibility
>
> >The paper does not explore the computational complexity of the model, nor does it discuss the feasibility of deploying SDTagNet in real-time systems with high computational constraints. The inference speed and memory requirements, especially with large-scale map data, are important factors for practical deployment, and they need to be addressed more thoroughly.
>
> SDTagNet is designed with efficiency in mind: the NLP tag encoder is based on a compact BERT model (embedding dim 144), and the SD map encoder is lightweight compared to the main perception backbone. We have included the parameter count and FPS of SDTagNet and the baselines on an NVIDIA H100 GPU below. The additional inference time for SDTagNet is less than 7 ms per frame, and the memory overhead is marginal relative to the base model.
> We thus consider our approach suitable for real-time deployment in autonomous driving systems, with a negligible difference to other SD map encoding methods. We will include these results in the final version of the Appendix.
>
> #### Table 7: Computational Complexity and Real-Time Feasibility
>
> | Model                   | Param. Count | FPS  | VRAM [MB] |
> |-------------------------|--------------|------|------|
> |    MapTRv2 (no prior)   |  42.12 M     | 16.0 |   3654  |
> |    MapTRv2 + PMapNet    |  50.48 M     | 13.9 |   3698  |
> |   MapTRv2 + SMERF       |  44.62 M     | 14.9 |   3662  |
> | MapTRv2 + SDTagNet      |  50.54 M     | 14.5 |   3686  |
>
> *Comparison of model parameter count, FPS, and VRAM with MapTRv2 and two SD map baseline methods. All experiments are on the Argoverse 2 dataset with far-range setting, averaged over 10 samples and tested on NVIDIA H100 GPU.*
>
> ### W5: Methodological Details: Processing of Complex Map Elements
>
> >While the introduction of the point-level SD map encoder is a key innovation, the methodology lacks detailed explanation on how it processes complex map elements (points, polylines, relations) and how it handles the integration of these diverse map features at a computational level.
>
> All map elements are encoded at the point level, resulting in identical processing steps, with orthogonal random feature (ORF) identifiers (think of IDs in vector space) enabling the model to distinguish and relate different element types.
> As depicted in Figure 4, points are represented as a single query of the SD map encoder with two identical ORF identifiers, a tag embedding, and positional encoding.
> Polylines are represented by a list of multiple point-like queries that all share the same two identical ORF identifiers. Relations are represented using pairs of ORF identifiers, each corresponding to one of the members of a relation edge, as done traditionally in graph-structured data with node IDs.
> We will highlight the commonality of processing and add further explanation to the camera-ready version.
>
> As mentioned in our reply to reviewer kgfW, substantial parts of the ablations done in Table 5 will be moved to the main paper. This ablation highlights the impact of ORF identifiers on detection performance and the symbiotic nature of combining it with the point-level representations and NLP-based tag embeddings.
>
> ---
>
> **Remark on Generalization to Diverse Geographic Regions**
>
> - **Generalization to Diverse Geographic Regions:**
>  We agree that further evaluation on more diverse regions would strengthen the work. Our use of geographically disjoint splits (Section 4.1) is a step towards realistic generalization assessment. We will extend our evaluation to additional regions and newly released datasets as they become available in future work.

---

> > ### Author Response · Authors · 2025-08-06
> >
> > We would like to highlight that we have added a new response to reviewer Xgo9 that compares SDTagNet with a wider range of baseline HD map construction models, further supporting the significance of the performance gains acheivable with our method. We evaluate SDTagNet on the ground truth of MapTracker - originally not chosen because it is easier and less realistic - and compare it with recent state-of-the-art works that do not use SD map priors. The results mirror our findings from the main evaluation, with SDTagNet still significantly outperforming all other approaches in the far range (+6.1 mAP vs. MapTracker), despite SDTagNet being the only single-shot method in this comparison other than HIMap.
> >
> > We would be grateful for any additional feedback you might have regarding our responses or if there are any remaining concerns that would benefit from further clarification.

---

### Official Review · Reviewer_kgfW · 2025-07-01

**Clarity:** 4
**Significance:** 3
**Originality:** 3
**Rating:** 4
**Confidence:** 5

**Summary:**

Existing methods for generating real-time high-definition (HD) maps for autonomous vehicles do not fully utilize the rich textual and structural data available in standard-definition (SD) maps like OpenStreetMap. The authors introduce SDTagNet, a novel approach that uses a natural language processing model to interpret textual annotations and a point-level encoder to uniformly integrate various map elements such as points, polylines, and their relationships. This method significantly enhances the accuracy of online HD map construction, demonstrating substantial performance gains on the Argoverse 2 and nuScenes datasets, especially at far perception ranges.

**Questions:**

I would like to increase my score if the authors can address my 1st and 3rd points in weaknesses

**Ethical Concerns:**

["NO or VERY MINOR ethics concerns only"]

**Final Justification:**

Based on the rebuttal and other reviewers' comment, I am maintaining my score.

**Limitations:**

Yes

**Quality:**

4

**Strengths And Weaknesses:**

Strengths:
1. This paper's key strength lies in its innovative use of textual annotations from Standard Definition (SD) maps, a rich source of information previously neglected in online HD map construction. The authors ingeniously employ a pre-trained BERT-based NLP encoder to interpret this vast and unstructured text data, eliminating the need for manual feature engineering and predefined class taxonomies that limited prior work.
2. This paper demonstrates a crucial strength in significantly boosting map perception performance at far ranges, a scenario where sensor data quality typically diminishes. This improvement is vital for autonomous driving, as accurate long-distance environmental models are necessary for safe operation at higher speeds.
3. This paper's move to a point-level query resolution for the SD map encoder is a significant architectural strength. This design choice creates better alignment with the underlying HD map construction model, which also operates on a per-point basis, and provides far greater flexibility and expressiveness than previous polyline-level methods.

Weaknesses:
1. This paper presents multiple significant improvements simultaneously, including a finer point-level query resolution and a novel method for leveraging textual tag annotations. While the authors rightly conduct ablation studies to parse the effectiveness of each individual component, this crucial analysis is relegated to an appendix. Placing this vital experimental breakdown in the appendix weakens the main paper, as it requires readers to consult supplementary materials to fully understand the specific contribution and importance of each innovation to the final impressive results
2. A notable limitation of this paper is the minimal performance improvement in the near-range setting, where the proposed method provides only a marginal gain over the baseline on both the Argoverse 2 and nuScenes datasets. This is likely because the high-quality sensor data available for the immediate vicinity is sufficient for the baseline model, reducing the reliance on SD map priors. Consequently, this indicates the primary utility of SDTagNet is to enhance perception where sensor data is sparse or unreliable
3. While the paper effectively demonstrates the benefits of its proposed method on the MapTRv2 architecture, its claims of efficacy could be strengthened by testing the SDTagNet module on a wider range of baseline HD map construction models.

---

> ### Author Rebuttal · Authors · 2025-07-30
>
> # Author Response to Reviewer kgfW
>
> We thank the reviewer for their valuable and constructive feedback and for recognizing the strengths of our work, including its main innovation of fully exploiting the rich information contained in SD maps, its architectural contributions, generalization capabilities, and significant improvement especially for far-range online HD map construction.
>
> ### W1: Placement of Ablation Studies
>
> > While the authors rightly conduct ablation studies to parse the effectiveness of each individual component, this crucial analysis is relegated to an appendix. Placing this vital experimental breakdown in the appendix weakens the main paper, as it requires readers to consult supplementary materials to fully understand the specific contribution and importance of each innovation to the final impressive results
>
> * In the camera-ready version, we will move as much of the main ablation results into the main paper as space allows and summarize the main findings. Additionally, we will reference explicitly to the appendix for further details, avoiding the full ablations to go unnoticed.
>
> ### W2: Near Range Performance
>
> > A notable limitation of this paper is the minimal performance improvement in the near-range setting, where the proposed method provides only a marginal gain over the baseline on both the Argoverse 2 and nuScenes datasets. This is likely because the high-quality sensor data available for the immediate vicinity is sufficient for the baseline model, reducing the reliance on SD map priors. Consequently, this indicates the primary utility of SDTagNet is to enhance perception where sensor data is sparse or unreliable.
>
> * Regarding the limited improvement in the near-range setting, we agree with the reviewer's analysis: the baseline MapTRv2 already performs strongly where sensor data is reliable, and the main benefit of SD maps is in far-range scenarios where areas are sparsely visible.
> * SD maps can still provide valuable prior knowledge even in near range. When we take a detailed look at the class-wise evaluation metrics we can conclude that based on the kind of information that SD maps typically provide.
> * For instance, detailed lane divider positions and marking types (dashed vs solid) are clearly visible in near range sensor data, thus SD map prior information contributes little in these cases or even slightly decreases the performance in the near range due to differing labeling guidelines between OSM and Argoverse 2.
> * But for classes such as lane centerlines, that have corresponding prior information in OSM, like upcoming turn directions (lane topology at intersections), lane count, road width, lane and road types, we see a notable improvement of the detection performance. Analogously, road boundary detections improve since OSM contains building outlines, pedestrian walkways, rough positions of roads, and their intersections, aiding the HD map construction in the near range as well.
>
> #### Table 4: Per-class breakdown of near range performance
>
> | Method | AP_dsh | AP_sol | AP_bou | AP_cen | AP_ped | mAP | vs. MapTRv2 |
> |--------|--------|--------|--------|--------|--------|-----|-------------|
> | MapTRv2 | 37.9 | 55.0 | 49.7 | 48.2 | 41.7 | 46.5 | \- |
> | \+ SDTagNet | 36.0 | 55.2 | **53.3** | **52.6** | 43.3 | 48.1 | **+1.6** |
> | Δ in AP | \-1.9 | +0.2 | **+3.6** | **+4.4** | +1.6 | 48.1 |  |
>
> *Per-class comparison of performance in the near range setting of MapTRv2 vs. SDTagNet. Values are taken from Table 1 in the paper.*
>
> ### W3: Generalization to Other Architectures:
>
> > While the paper effectively demonstrates the benefits of its proposed method on the MapTRv2 architecture, its claims of efficacy could be strengthened by testing the SDTagNet module on a wider range of baseline HD map construction models.
>
> * We evaluate against two prominent SD map prior baselines, SMERF and PMapNet, both re-implemented and integrated on top of the same single-shot HD map construction model for fair comparison. This ensures consistent evaluation conditions and isolates the effects of the SD map prior encoders.
> * Rather than comparing reported numbers across incompatible setups, we take a more rigorous approach by rebuilding prior methods on a unified baseline that is significantly stronger than the original base architectures of SMERF and PMapNet.
> * We selected MapTRv2 as the base model because it is a widely adopted and strong baseline for online HD map construction. Many recent methods build on its foundation, reinforcing our choice to use it for isolating the effect of SD map priors in a stable setup.
> * Other recent work like HIMap \[h\], MapUnveiler \[i\], and MapTracker \[c\] contains relevant advances in their setting, but either has no public code available, or, in the case of MapTracker, introduce complexity for effective integration hat we consider outside the scope of this paper. A more detailed discussion of these works can also be found in the response to reviewer Xgo9, who shared similar feedback.
>
> \[b\] Chen, Jiacheng, et al. "Maptracker: Tracking with strided memory fusion for consistent vector hd mapping." ECCV, 2024.
>
> \[d\] Luo, Katie Z et al. "Augmenting lane perception and topology understanding with standard definition navigation maps.", ICRA, 2024
>
> \[e\] Jiang, Zhou et al. "P-mapnet: Far-seeing map generator enhanced by both sdmap and hdmap priors." RA-L, 2024
>
> \[h\] Zhou, Yi, et al. "Himap: Hybrid representation learning for end-to-end vectorized hd map construction." CVPR, 2024.
>
> \[i\] Kim, Nayeon, et al. "Unveiling the Hidden: Online Vectorized HD Map Construction with Clip-Level Token Interaction and Propagation." NeurIPS, 2024.

---

> > ### Author Response · Authors · 2025-08-06
> >
> > We would like to highlight that we have added a new response to reviewer Xgo9 that compares SDTagNet with a wider range of baseline HD map construction models. We evaluate SDTagNet on the ground truth of MapTracker - originally not chosen because it is easier and less realistic - and compare it with recent state-of-the-art works that do not use SD map priors. The results mirror our findings from the main evaluation, with SDTagNet still significantly outperforming all other approaches in the far range (+6.1 mAP vs. MapTracker), despite SDTagNet being the only single-shot method in this comparison other than HIMap.
> >
> > We would be grateful for any additional feedback you might have regarding our responses or if there are any remaining concerns that would benefit from further clarification.

---

### Official Review · Reviewer_Xgo9 · 2025-07-02

**Clarity:** 3
**Significance:** 3
**Originality:** 3
**Rating:** 4
**Confidence:** 5

**Summary:**

The paper proposes an approach for online high-definition (HD) map construction that leverages the rich semantic and textual information available in standard-definition (SD) maps like OpenStreetMap, overcoming the limitations of previous methods that only used a restricted subset of SD map features. By introducing a BERT-based natural language processing (NLP) encoder for open-vocabulary tag embeddings and a point-level SD map encoder with orthogonal element identifiers, SDTagNet can uniformly integrate points, polylines, and relations, thus eliminating the need for manual feature engineering or predefined class taxonomies. This architecture enables the model to utilize all available semantic information, including diverse textual annotations, and supports scalable, self-supervised pre-training on large-scale SD map data, making it robust and adaptable for real-world autonomous driving scenarios. Experimental results on the Argoverse 2 and nuScenes datasets demonstrate that SDTagNet improves far range HD map construction performance, achieving state-of-the-art performance.

**Questions:**

The idea proposed in this paper is interesting, as it makes full use of practically available information from SD maps, which is highly relevant for real-world applications. I expect this approach will inspire many related follow-up studies. However, my main concern lies with the experimental setup. Since the method leverages additional information from SD maps, it would be expected that the proposed method outperforms existing HD map construction approaches. Additionally, I think the proposed method would be performed on top of various baselines, not only on top of MapTRv2. While I currently stand positive, my score could be changed if my concerns are not addressed.

**Ethical Concerns:**

["NO or VERY MINOR ethics concerns only"]

**Final Justification:**

My concerns have been adequately addressed through the authors' feedback. Thus, I will maintain my initial positive rating.

**Limitations:**

yes

**Paper Formatting Concerns:**

I have no paper formatting concerns.

**Quality:**

3

**Strengths And Weaknesses:**

**Strengths**
1. The idea of leveraging text-annotated standard-definition (SD) maps is highly innovative, offering a promising approach that is well-suited for real-world deployment in autonomous driving scenarios.

2. The overall model design is well-structured and logically justified, integrating natural language processing for tag embeddings and a flexible SD map encoder to effectively utilize diverse map elements.

3. Experimental results demonstrate that the proposed method significantly outperforms previous approaches that utilize SD maps, achieving superior performance in online high-definition (HD) map construction tasks.

4. The code submission enhances the reproducibility and trustworthiness of the experimental results, allowing other researchers to verify and build upon the work more effectively.

**Weaknesses**
1. The comparison table in the paper is incomplete, as it does not include several recent and relevant methods for online vectorized HD map construction such as HIMap [a], MapUnveiler [b], and MapTracker [c]; since the proposed method additionally leverages SD maps, it is important to directly compare its performance against these state-of-the-art approaches to robustly validate its effectiveness.

2. The proposed method is evaluated only on the geographical split of [17], whereas many prior works opt for StreamMapNet split [d]; conducting experiments on this commonly used split would enable fair and comprehensive comparisons with a broader range of existing methods and further strengthen the empirical validation of SDTagNet. Furthermore, the author may consider using a consistent GT used in MapTracker [c] for a better thorough experiment.

[a] Zhou, Yi, et al. "Himap: Hybrid representation learning for end-to-end vectorized hd map construction." CVPR, 2024.\
[b] Kim, Nayeon, et al. "Unveiling the Hidden: Online Vectorized HD Map Construction with Clip-Level Token Interaction and Propagation." NeurIPS, 2024.\
[c] Chen, Jiacheng, et al. "Maptracker: Tracking with strided memory fusion for consistent vector hd mapping." ECCV, 2024.\
[d] Yuan, Tianyuan, et al. "Streammapnet: Streaming mapping network for vectorized online hd map construction." WACV, 2024.

---

> ### Author Rebuttal · Authors · 2025-07-30
>
> # Author Response to Reviewer Xgo9
>
> We thank the reviewer for their valuable feedback and for recognizing the strengths of our work, including its main innovation of using language processing, the flexibility of our SD map encoding, and our contribution towards reproducibility. We are encouraged by the reviewer's assessment that our approach 'will inspire many related follow-up studies,' which validates the significance of our contribution.
>
> ### W1: Comparison to Recent Methods (HIMap, MapUnveiler, MapTracker)
>
> > The comparison table in the paper is incomplete, as it does not include several recent and relevant methods for online vectorized HD map construction such as HIMap [a], MapUnveiler [b], and MapTracker [c]; since the proposed method additionally leverages SD maps, it is important to directly compare its performance against these state-of-the-art approaches to robustly validate its effectiveness.
>
> and
>
> >Since the method leverages additional information from SD maps, it would be expected that the proposed method outperforms existing HD map construction approaches.
>
>
> We appreciate the suggestion to include these recent methods and agree they present relevant advances in their setting.
>
> - MapTracker \[b\] (and its predecessor StreamMapNet \[a\]) marks a major shift in HD map construction through its use of end-to-end temporal modeling (MOTR-based tracking). This complements our direction, which focuses on leveraging static SD maps—a distinct and orthogonal source of information. SD map priors aim to enhance map construction performance even in previously unseen areas, including those far into the future range. These two mechanisms could potentially complement each other in the future.
> - Simply comparing the reported results of HIMap, MapUnveiler, or MapTracker is not feasible due to inconsistent validation splits and ground truth label sets. Each paper applies different HD map references or simplified label spaces (e.g., 3-class vs. 5-class).
> - For HIMap and MapUnveiler sadly no public code is available (git repositories are empty), which we would require to incorporate SDTagNet and evaluate them on our expanded and consistent ground truth. We have contacted the authors of HIMap regarding the code while working on SDTagNet but unfortunately did not receive a response.
> - In particular, integrating SDTagNet with temporally-enhanced architectures such as MapTracker presents promising future work. However, as shown in our ablations, effective integration of SD priors requires careful architectural design. MapTracker’s multi-stage, multi-loss, memory-bank-based system introduces complexity that we consider outside the scope of this paper.
>
> > Additionally, I think the proposed method would be performed on top of various baselines, not only on top of MapTRv2.
> >
>
> - We evaluate against two prominent SD map prior baselines, SMERF and PMapNet, both re-implemented and integrated on top of the same single-shot HD map construction model for fair comparison. This ensures consistent evaluation conditions and isolates the effects of the SD map prior encoders.
> -  Rather than comparing reported numbers across incompatible setups, we take a more rigorous approach by rebuilding prior methods on a unified baseline that is significantly stronger than the original base architectures of SMERF and PMapNet.
> - We selected MapTRv2 as the base model because it is a widely adopted and strong baseline for online HD map construction. Many recent methods, including HIMap, build on its foundation, reinforcing our choice to use it for isolating the effect of SD map priors in a stable setup.
>
> ### W2: Evaluation on StreamMapNet Split and Consistent GT
>
> >The proposed method is evaluated only on the geographical split of [17], whereas many prior works opt for StreamMapNet split [d]; conducting experiments on this commonly used split would enable fair and comprehensive comparisons with a broader range of existing methods and further strengthen the empirical validation of SDTagNet. Furthermore, the author may consider using a consistent GT used in MapTracker [c] for a better thorough experiment.
>
> **Response:**
> - We chose the geographically non-overlapping split from [17] (Lilja et al., CVPR 2024) to avoid overfitting to specific locations, which is especially important when using map priors. As shown in our experiments (see supplementary material, Table 6), models using SD map priors are more prone to geographic overfitting compared to models without any prior. Additionally, the geo split provides a more realistic assessment of generalization to unseen areas.
>
> - The StreamMapNet split is geographically disjoint as well. At the time we started this work, it was not clear whether the split from [17] or the StreamMapNet split would prevail. To facilitate comparisons, we provide results on both splits, showing that our method maintains strong performance in either setting. The performance of all baselines and our method trained on the StreamMapNet split can be found in Table 2.
>
> #### Table 2: StreamMapNet Split
>
> | Model                                | AP_div | AP_sol | AP_bou | AP_cen | AP_ped | **mAP**    | Δ mAP to no prior |
> |--------------------------------------|--------|--------|--------|--------|--------|------------|-------|
> |    MapTRv2 (no prior)                | 5.5 | 14.5 | 12.9 | 16.3 | 16.3 | **13.1** | 0.0       |
> |    MapTRv2 + PMapNet                 | 6.6 | 15.3 | 16.4 | 19.1 | 20.8 | **15.6** | +2.5      |
> |   MapTRv2 + SMERF                    | 5.6 | 14.8 | 17.0 | 18.4 | 16.0 | **14.4** | + 1.3     |
> | MapTRv2 + SDTagNet                   | 7.8 | 19.7 | 24.7 | 24.0 | 28.2 | **20.9** | **+7.8**      |
>
> *Evaluation on Argoverse 2 dataset with StreamMapNet geographical split, far range (120m × 60m) setting, 6 epochs training.*
>
> - We agree that consistent ground truth is essential for thorough evaluation. As noted in Section 4.1, SDTagNet uses the improved labels from M3TR \[g\], which to our knowledge is the most comprehensive and most challenging ground truth. They include topologically relevant centerlines and correct the same errors in the original labels as Maptracker did for the consistent GT. Table 3, reproduced from M3TR, provides an overview of proposed ground truths of different online HD map construction approaches.
>
> #### Table 3: Comparison of Ground Truth Labels
>
> | **Method**                | Divider Types | Lane Centerlines | 3D Instances | Fixed GT Artifacts | Geo. Split |
> |---------------------------|:-------------:|:-------------:|:-------------:|:------------------:|:----------:|
> | VectorMapNet          |      -        |      -        |      -        |        -           |     -      |
> | MapTRv2           |      -        |   ✓           |   ✓           |        -           |     -      |
> | StreamMapNet         |      -        |      -        |      -        |        -           |   ✓        |
> | MapTracker            |      -        |      -        |      -        |      ✓             |   ✓        |
> | MapEX                 |      -        |      -        |      -        |        -           |     -      |
> | PriorDrive            |      -        |      -        |      -        |        -           |     -      |
> | **M3TR**      |   ✓           |   ✓           |   ✓           |      ✓             |   ✓        |
>
> ---
>
> ### References
>
> \[a\] Yuan, Tianyuan, et al. "Streammapnet: Streaming mapping network for vectorized online hd map construction." WACV, 2024.
>
> \[b\] Chen, Jiacheng, et al. "Maptracker: Tracking with strided memory fusion for consistent vector hd mapping." ECCV, 2024.
>
> \[c\] Liao, Bencheng, et al. "MapTRv2: An End-to-End Framework for Online Vectorized HD Map Construction" IJCV, 2024
>
> \[d\] Luo, Katie Z et al. "Augmenting lane perception and topology understanding with standard definition navigation maps.", ICRA, 2024
>
> \[e\] Jiang, Zhou et al. "P-mapnet: Far-seeing map generator enhanced by both sdmap and hdmap priors." RA-L, 2024
>
> \[g\] Immel et al. *M3TR: A Generalist Model for Real-World HD Map Completion.* arxiv, 2025.
>
> ## Additional Notes
>
> We have added our code to the submission and will release it along with pre-trained models upon publication. We aim to promote reproducibility and encourage further innovations in SD map prior integration—potentially also in conjunction with temporal models like Maptracker.
>
>
> We would like to thank the reviewer again for their positive assessment and valuable suggestions. We will incorporate our additional clarifications and evaluations in the final version.

---

> ### Comment · Reviewer_Xgo9 · 2025-08-01
> **Response to the rebuttal**
>
> Thank you for the rebuttal. My main concern about the lack of direct comparison to state-of-the-art methods still stands. All current experiments rely on in-house reproductions rather than comparing to published papers, and the reported mAP scores are very low (mAP < 50).\
> I had suggested using standard benchmarks like the StreamMapNet split [d] or MapTracker GT [c] and compare with state-of-the-art approaches such as HIMap [a], MapUnveiler [b], and MapTracker [c] for a fair comparison. The new table provided in the rebuttal, however, uses a non-standard 120m×60m range, which prevents direct comparison with other papers. Additionally, the reported mAP of 20.9 is also too low to be considered practical.\
> Based on MapTracker paper [c], MapTRv2 achieves an mAP of 70.9 on the standard StreamMapNet split [d] (60m×30m range), and even the StreamMapNet (2018) achieves 54.6 mAP (100m×50m range+MapTracker GT [c]). Given my experience, my concern remains that the current experimental setup may have been chosen to highlight the method's relative strengths, rather than demonstrating its absolute, practical significance. It is still unclear whether the proposed method offers a meaningful advancement over existing techniques or if the results are a product of experimental cherry-picking.

---

> > ### Author Response · Authors · 2025-08-04
> >
> > > It is still unclear whether the proposed method offers a meaningful advancement over existing techniques or if the results are a product of experimental cherry-picking.
> >
> > - We understand that concern. However, if we wanted to exploit absolute numbers, we would not have adapted the other SD map approaches' base architectures to the significantly more powerful MapTRv2, but simply compared their numbers, hiding which benefit comes from our SD map innovation vs a better base architecture. Analogously, we could have easily chosen easier Chamfer distance thresholds to boost our numbers, but consciously chose the same hard thresholds that are applied in the near range. In the rebuttals, we provided additional numbers with those easier Chamfer distance thresholds (see Table 1).
> >
> > > and the reported mAP scores are very low (mAP < 50).
> >
> > - As noted in our response to reviewer iYXD, absolute mAP scores are strongly dependent on the chosen dataset split, label set and additional parameters like 2D vs. 3D instances.
> > - The geographic split paper of [17] (Lilja et al., CVPR 2024), which we used for our evaluation split, reports the results for MapTRv2 and StreamMapNet (among others) on the original and their new split on Argoverse 2, which we have included below for convenience. Considering the fact that our ground truth setting is significantly harder, with 3 instead of 5 classes and 3D instances, SDTagNet maintains expected performance when compared with the same dataset split.
> >
> > | Method | Split (near range) | Divider | Boundary | Crossing | Mean |
> > |---------|----------|-------|-------|-------|-------|
> > | VectorMapNet 2D | AV2 Original Split | 51.9 | 42.1 | 38.0 | **44.0** |
> > | | AV2 Geo Split | 39.8 | 31.5 | 26.8 | **32.7** |
> > | MapTR 2D | AV2 Original Split | 64.0 | 63.2 | 63.7 | **63.6** |
> > | | AV2 Geo Split | 50.0 | 47.5 | 46.6 | **48.0** |
> > | MapTRv2  2D | AV2 Original Split | 71.7 |  67.0 | 64.5 | **67.7** |
> > | | AV2 Geo Split | 58.4 | 51.3 | 49.7 | **53.1** |
> > | MapTRv2  3D | AV2 Original Split | 68.7 | 64.3 | 59.6 | **64.2** |
> > | | AV2 Geo Split | 56.2 | 47.8 | 46.2 | **50.1** |
> > | StreamMapNet 2D | AV2 Original Split | 58.3 | 63.9 | 62.7 | **61.7** |
> > | | AV2 Geo Split | 52.7 | 50.0 | 49.4 | **50.7** |
> >
> > > Based on MapTracker paper [c], MapTRv2 achieves an mAP of 70.9 on the standard StreamMapNet split [d] (60m×30m range),
> >
> > We would like to highlight that this value from the MapTracker paper is for the original (non-geo) split of Argoverse 2 and with 2D instead of 3D instances. In Table 6 in the supplementary material we have reported the results of MapTRv2 and SDTagNet trained on the original AV2 split.
> >
> > > uses a non-standard 120m×60m range, which prevents direct comparison with other papers.
> >
> > The 120x60m range is not our own or chosen to optimize our results but that used by PMapNet [e], enabling direct comparison with one of the two existing SDMap prior baselines that we evaluated against (SMERF only evaluates in near range).
> >
> > > I had suggested using standard benchmarks like the StreamMapNet split [d] or MapTracker GT [c]
> >
> > We have now also started a training on MapTracker's labels, which - in our view - are less realistic, easier, and only predicted in 2D (instead of 3D). We will provide more evaluation results are soon as they roll in.

---

> ### Author Response · Authors · 2025-08-06
>
> We have now completed the training of SDTagNet on the ground truth setting of MapTracker, the results of which are collected in the table below. We have trained 4 variants - near range and far range, respectively with and without the StreamMapNet geo-split (which is also used by Maptracker). The table additionally contains the results of the discussed recent state-of-the-art methods where available from their publications. We will include this additional experiment in the final version.
>
> The results mirror our findings from the main evaluation, with SDTagNet still significantly outperforming all other approaches in the far range (**+6.1 mAP vs. MapTracker**), despite SDTagNet being the only single-shot method in this comparison other than HIMap.
> We believe this further demonstrates the meaningful advancement over existing techniques, especially as the SDTagNet module could be combined with temporal tracking similar to MapTracker for even larger performance gains.
>
>
> | Method          | Dataset + Split          | Range     | Epochs | Divider | Boundary | Crossing |     Mean |
> |-----------------|--------------------------|-----------|--------|---------|----------|----------|----------|
> | HIMap [h]       | AV2 Original Split       | 60m x 30m |**24**  | 72.4    |  73.2    | 72.4     | **72.7** |
> | MapUnveiler [i] | AV2 Original Split       | 60m x 30m |  30    | 74.2    |  71.9    | 72.5     | **72.9** |
> | StreamMapNet** [a]| AV2 Original Split       | 60m x 30m |  72    | 74.2    |  66.1    | 70.5     | **70.3** |
> | MapTracker [b]  | AV2 Original Split       | 60m x 30m |  35    | 80.0    | 73.7     | 77.0     | **76.9** |
> | **SDTagNet**    | AV2 Original Split       | 60m x 30m |**24**  |  81.7   |     76.3 | 76.1     | **78.0** |
> |                 |                          |           |        |         |          |          |          |
> | StreamMapNet** [a]|AV2 StreamMapNet Geo Split| 60m x 30m |  72    | 68.2    |  63.2    | 61.8     | **64.4** |
> | MapTracker [b]  |AV2 StreamMapNet Geo Split| 60m x 30m |  35    | 75.1    |  68.9    | 70.0     | **71.3** |
> | **SDTagNet**    |AV2 StreamMapNet Geo Split| 60m x 30m |**24**  | 72.0    | 67.5     | 64.0     | **67.8** |
> |                 |                          |           |        |         |          |          |          |
> | MapUnveiler [i] | AV2 Original Split       | 100m x 50m|  30    | 67.9    | 62.6     | 71.7     | **67.4** |
> | StreamMapNet [a]| AV2 Original Split       | 100m x 50m|  30    |-*       |-*        |-*        | **57.7** |
> | **SDTagNet**    | AV2 Original Split       | 100m x 50m|**24**  | 80.2    | 72.9     | 81.7     | **78.3** |
> |                 |                          |           |        |         |          |          |          |
> | StreamMapNet** [a]|AV2 StreamMapNet Geo Split| 100m x 50m|  72    | 56.1    |  47.5    | 60.1     |**54.6**  |
> | MapTracker [b]  |AV2 StreamMapNet Geo Split| 100m x 50m|  35    | 64.6    |  58.5    | 71.2     |**64.8**  |
> | **SDTagNet**    |AV2 StreamMapNet Geo Split| 100m x 50m|**24**  | 69.5    |  68.0    | 75.1     | **70.9** |
>
> *\* = only total mAP reported in the original paper*
>
> *\*\* = values taken from MapTracker [b] paper*
>
> *following StreamMapNet [a], MapTracker [b] and MapUnveiler [i], near range mAP is calculated with thresholds of  [0.5m, 1m, 1.5m] and far range mAP with thresholds of [1m, 1.5m, 2m]*

---

### Official Review · Reviewer_iYXD · 2025-07-02

**Clarity:** 2
**Significance:** 2
**Originality:** 2
**Rating:** 4
**Confidence:** 3

**Summary:**

This paper proposes a method to incorporate map data from a standard-definition map data set like OSM into a high-definition map-generation pipeline (where a high-definition map includes detailed road and lane boundary information). The key technical contributions are:

- A way to incorporate natural language road "tags" that are common in OSM, such as annotations about the number of lanes or road types, based on an NLP (BERT) encoder trained with a contrastive loss metric to predict if two annotations are “semantically similar,” i.e., they encode a road segment with the same number of lanes and the same direction of travel.

- An overall encoding of SD map data that includes these semantic embeddings and also differs from prior work by encoding road segments as points instead of polylines and a method for encoding relational features (such as the fact that two points belong to the same segment).

Evaluation is done on the nuScenes and Argoverse datasets, showing that the method can improve over a method that relies only on a method that does not use SD map features (MapTRv2), and that it generally outperforms existing methods that add SD map features to the HD map generation.

**Questions:**

I’d like to see the authors response to W1/W2/W4.

For W1/W2, given the overall low MAP and small absolute gains over MapTRv2, is SD map augmentation really an important component forsolving the HD Map generation problem?

**Ethical Concerns:**

["NO or VERY MINOR ethics concerns only"]

**Final Justification:**

Based on the reviewers comments and my reading of the other reviews, I stand by my original assessment.

**Limitations:**

The authors do a good job of describing limitations, which illustrates qualitative examples where the method doesn't work, and in the body of the text where they talk about why the gains over the baseline are not huge.

**Quality:**

3

**Strengths And Weaknesses:**

Strengths:
S1 Clever method for encoding semantic tags and other road metadata.
S2 Overall SD map encoding also seems quite general and flexible and appears to be better than competing methods.
S3 Evaluation against state-of-the-art baselines on real-world data sets shows some gains.
S4 Paper and supplementary material do a good job of listing weaknesses and comparing against a number of alternatives.

Weaknesses:
W1 Overall gains are modest; in the “near range” setting, the method helps only marginally, in the “far range” setting it does somewhat better, but the overall accuracy in this setting is quite low (so it’s unclear if the gains are really meaningful from the point of view of, e.g., autonomous vehicle navigation).

W2 While I appreciate the methods that the authors have developed, I found it hard to be excited about the problem here — overall the contribution is modest, and I’m not sure whether the whole line of “add SD Map Data to HD Map data” is going to ultimately deliver the types of significant accuracy gains that are needed to solve the HD map generation problem.

W3 I found the paper a bit hard to follow in places. For example, Figure 1 shows that images are used to produce BEV features, but the paper doesn’t really talk about this part of the architecture at all. It wasn’t until I got to the experiments that I realized that the image features are all based on a pre-trained image/lidar to HD Map model (MapTRv2). In other places, there is jargon — for example, “BEV” is used without being defined, and phrases like “we also resample all SD map polylines to the fixed point number of 10” don’t mean anything to me.

W4 I don’t quite follow why the baselines, especially PMapNet, don’t seem to show gains in this evaluation. On nuScenes, the PMapNet paper suggests that it offers significant performance boosts vs. HDMapNet — is it just that MapTRv2 is a much stronger baseline than HDMapNet? Why do you think it is that the (all info) baselines don’t really do much better than the originals?

---

> ### Author Rebuttal · Authors · 2025-07-30
>
> # Author Response to Reviewer iYXD
>
> We thank the reviewer for their thoughtful and constructive feedback, as well as for recognizing the strengths of our work, including the generality and flexibility of our SD map encoding, the novel semantic tag embedding, the thorough evaluation, and discussion of limitations.
>
> ## Response to Specific Weaknesses
>
> ### W1: Modest Gains in Near Range, Low Absolute Accuracy in Far Range
>
> > Overall gains are modest; in the “near range” setting, the method helps only marginally, in the “far range” setting it does somewhat better, but the overall accuracy in this setting is quite low (so it’s unclear if the gains are really meaningful from the point of view of, e.g., autonomous vehicle navigation).
>
> We appreciate the reviewer’s concern regarding the significance of the reported gains, particularly in the far range setting where absolute mAP values are low. We would like to clarify several points:
>
> - As Reviewer kgfW also noted, for the near range scenario, the high-quality sensor data in the immediate vicinity is likely sufficient information for some element classes. However SDTagNet still shows significant gains for road boundaries and lane centerlines, even in the near range. The Response to Reviewer kgfW contains a detailed analysis of the performance difference per element class. The far range scenario in contrast is inherently more challenging due to limited sensor visibility, occlusions, and lower image resolution of visible far-range areas.
>
>
> - The low absolute mAP in the far range is also an artifact of our chosen evaluation metric. The mAP in online HD map construction is usually calculated via the averaging of Chamfer distance at specific thresholds, commonly \[0.5m, 1m, 1.5m\] in the near range setting. We adopted the same thresholds for both near and far range setting to make the values directly comparable and show the benefits of SDTagNet in a hard evaluation context.
>
> - This however causes the absolute mAP values to appear quite low. For comparison, StreamMapNet [a], MapTracker [b], and other approaches use higher thresholds of \[1m, 1.5m, 2m\] in their far range setting. To make SDTagNet comparable with their evaluations, we provide the mAP values for Argoverse 2 in far range with higher thresholds in the table below. The absolute mAP values of all models increase drastically, with the performance gain of SDTagNet increasing proportionally, totaling +10.1 mAP vs. no prior and +6.4 mAP vs. the best other SD map prior encoding method.
>
> #### Table 1: Far range with higher mAP thresholds
>
> | Model | AP_div | AP_sol | AP_bou | AP_cen | AP_ped | **mAP** | vs. MapTRv2 |
> |-------|--------|--------|--------|--------|--------|---------|-------------|
> | MapTRv2 (no prior) | 14.1 | 23.3 | 23.5 | 27.3 | 27.0 | **23.0** | - |
> | MapTRv2 + PMapNet  | 14.3 | 25.2 | 27.8 | 30.4 | 35.6 | **26.7** | +3.7 |
> | MapTRv2 + SMERF    | 13.3 | 21.9 | 22.7 | 25.8 | 26.5 | **22.0** | -1.0 |
> | MapTRv2 + SDTagNet | 19.0 | 28.7 | 36.2 | 36.5 | 44.9 | **33.1** | **+10.1** |
>
> *Evaluation on Argoverse 2 dataset with geographical split, far range (120m × 60m) setting, higher mAP thresholds \[1m, 1.5m, 2m\] (like [a, b]).*
>
> ### W2: Significance of SD Map Priors for HD Map Generation
>
> > While I appreciate the methods that the authors have developed, I found it hard to be excited about the problem here — overall the contribution is modest, and I’m not sure whether the whole line of “add SD Map Data to HD Map data” is going to ultimately deliver the types of significant accuracy gains that are needed to solve the HD map generation problem.
>
> - SD maps (e.g., OSM) are globally available, change much less often and are significantly easier to maintain (both compared to HD maps). This makes them a practical prior in many use cases. The flexible, task-agnostic design and pre-training of the encoder complements this, making it suitable for applications beyond online HD map construction that also benefit from map context, such as trajectory prediction.
>
> - As we show above, absolute mAP values and gains strongly depend on the specific evaluation setting. When using easier thresholds that are commonly used \[a, b\] and correspond to the navigational flexibility at greater range, absolute mAP values increase drastically.
>
> - SDTagNet is furthermore able to correctly predict elements that are not visible in sensor data, for example crossing topologies, which we show in our qualitative examples. The accuracy gains in these settings are not achievable with other approaches, even other SD map encoders that do not incorporate semantic tags. This makes SDTagNet an important part in solving the HD map generation problem.
>
> ### W4: Baseline Performance, Especially PMapNet
>
> > I don’t quite follow why the baselines, especially PMapNet, don’t seem to show gains in this evaluation. On nuScenes, the PMapNet paper suggests that it offers significant performance boosts vs. HDMapNet — is it just that MapTRv2 is a much stronger baseline than HDMapNet? Why do you think it is that the (all info) baselines don’t really do much better than the originals?
>
> - MapTRv2 is indeed a much stronger baseline than HDMapNet. In the original MapTRv2 [c] paper, the authors report a mAP of 18.8 for HDMapNet and 67.4 for MapTRv2 on Argoverse 2. The evaluation in MapTRv2 was not performed on a geographic split (roads from the training set also appear in the validation set), and with 3 instead of 5 classes, so the values are not directly comparable to ours. However, the mAP difference should give an indication of baseline model performance strength.
> - Furthermore, PMapNet does not use a geographic split between train and val data like SDTagNet does, which enables overfitting on specific areas and roads - a problem which increases with the usage of priors like SD maps in a model.
> - The fact that the (all info) baselines don’t really do much better than the originals is consistent with our findings in the ablation studies of SDTagNet (Table 3 in the Appendix).
> We observed that simply "dumping" all possible information into the encoder is not enough, the encoder architecture plays a big role in model performance. For example, the inclusion of ORF element identifiers, which none of the competing encoders do, is very important for mAP results.
> Without the combination of all proposed architectural changes in SDTagNet, no baseline method is able to provide a significant performance gain over MapTRv2 in the near range on Argoverse 2.
>
> [a] Yuan, Tianyuan, et al. "Streammapnet: Streaming mapping network for vectorized online hd map construction." WACV, 2024.
>
> [b] Chen, Jiacheng, et al. "Maptracker: Tracking with strided memory fusion for consistent vector hd mapping." ECCV, 2024.
>
> [c] Liao, Bencheng, et al. "MapTRv2: An End-to-End Framework for Online Vectorized HD Map Construction" IJCV, 2024

---

> > ### Comment · Reviewer_iYXD · 2025-08-01
> >
> > Thank you for your response to my review.   Overall I stand by my initial evaluation -- I find the method described here to be interesting, but it's hard to tell if the gains are really significant.  In particular I'm swayed by the argument of reviewer Xgo9 -- it feels like the work in this area has not used a consistent set of baselines, and I'm bothered that small variations in the evaluation settings (such as the choice of chamfer distances in the table above) can have such a significant effect on the evaluation results.

---

> > > ### Author Response · Authors · 2025-08-06
> > >
> > > Thank you for maintaining this dialogue and for the continued positive evaluation. We would like to highlight that we have added a new response to reviewer Xgo9 that addresses their mentioned concerns. We evaluate SDTagNet on the ground truth setting of MapTracker - originally not chosen because it is easier and less realistic - and compare it with recent state-of-the-art works that do not use SD map priors. The results mirror our findings from the main evaluation, with SDTagNet still significantly outperforming all other approaches in the far range (+6.1 mAP vs. MapTracker), despite SDTagNet being the only single-shot method in this comparison other than HIMap. We believe that these additional baselines further support the significance of the gains achievable with our method.

---

### Decision · Program_Chairs · 2025-09-17

**Decision:**

Accept (poster)

**Comment:**

This paper presents an online high-definition map reconstruction method called SDTagNet, which utilizes text-annotated standard-definition maps such as OpenStreetMap as prior knowledge. The authors introduce a natural language processing model to interpret textual annotations and a point-level SD map element encoder to uniformly integrate various map elements. Extensive evaluations on the Argoverse 2 and nuScenes datasets confirm that the method achieves gains in map perception accuracy.

The paper received four “Borderline Accept” ratings.

Reviewer iYXD and Reviewer Xgo9 had concerns about the performance of SDTagNet and baseline methods. They both finally gave the "Borderline Accept". However, the authors further presented a detailed reply about these concerns.

Reviewer kgfW maintained the  “Borderline Accept” rating after reading the rebuttals and other reviewers' comments.

Reviewer CvWw thought the authors had processed all his/her concerns. Reviewer CvWw finally gave the "Borderline Accept".

The majority of the reviewers' concerns had been addressed. There is a consensus on the positive evaluation. This manuscript is recommended for acceptance.